# Dendritic cell entry to lymphatic capillaries is orchestrated by CD44 and the hyaluronan glycocalyx

Louise A Johnson[1], Suneale Banerji[1], B Christoffer Lagerholm[2], David G Jackson[1]

**DCs play a vital role in immunity by conveying antigens from peripheral tissues to draining lymph nodes, through afferent lymphatic vessels. Critical to the process is initial docking to the lymphatic endothelial receptor LYVE-1 via its ligand hyaluronan on the DC surface. How this relatively weak binding polymer is configured for specific adhesion to LYVE-1, however, is unknown. Here, we show that hyaluronan is anchored and spatially organized into a 400–500 nm dense glycocalyx by the leukocyte receptor CD44. Using gene knockout and by modulating CD44-hyaluronan interactions with monoclonal antibodies in vitro and in a mouse model of oxazolone-induced skin inflammation, we demonstrate that CD44 is required for DC adhesion and transmigration across lymphatic endothelium. In addition, we present evidence that CD44 can dynamically control the density of the hyaluronan glycocalyx, regulating the efficiency of DC trafficking to lymph nodes. Our findings define a previously unrecognized role for CD44 in lymphatic trafficking and highlight the importance of the CD44:HA:LYVE-1 axis in its regulation.**

## Introduction

Mobilization of antigen-presenting cells such as DCs from peripheral tissue via afferent lymphatics to draining LNs is essential for immune surveillance and for generating and regulating cellular immune responses (reviewed in reference 1). The process involves a number of individual steps guided by specific adhesion receptors and chemotactic cues, beginning with interstitial migration towards initial lymphatic capillaries, and followed by transit across endothelium to the vessel lumen and intralumenal crawling (2, 3, 4, 5, 6, 7, 8, 9, 10). A critical step in this process is that of vessel entry, during which migrating leukocytes with a diameter of ~10 $\mu$m must squeeze between the interdigitating endothelial flaps that comprise the distinctive overlapping junctions of initial lymphatic capillaries. These flaps are buttoned at their sides by the adherens junction molecule vascular endothelial cadherin (VE-cadherin) and tight junctional adhesion molecules including claudins and zonula

occludens-1 (ZO-1), and lined at their tips by the lymphatic endothelial hyaluronan (HA) receptor LYVE-1 and CD31 (11). From the results of targeted gene deletion and antibody blockade studies, it is now clear that entry at such junctions involves formation of dynamic LYVE-1-rich transmigratory cups, which permit docking of DCs via HA present as a dense layer on their surface (12).

HA is a ubiquitously expressed glycosaminoglycan composed of linear repeating units of $D$-glucuronic acid and $N$-acetyl-$D$-glucosamine (GlcNAc $_{\beta1-4}$ GlcUA)$_n$, which may extend to up to 20,000 disaccharide units in length, with contour lengths of up to 25 $\mu$m. HA is implicated in a wide range of physiological and pathological roles, as a structural component and also through highly regulated interactions with a cohort of HA-binding proteins. Generated endogenously by members of the HA synthase protein family located on the inner leaflet of the plasma membrane (reviewed in reference 13), the nascent polymers are extruded from the cell surface. However, the molecular details of the DC HA glycocalyx and its structural organization remain poorly defined. Most importantly, the supramolecular organization of large multivalent HA polymers within the DC glycocalyx provides the necessary avidity for efficient binding to LYVE-1, enabling the receptor to discriminate between HA-coated DCs and smaller uncomplexed HA molecules that bind LYVE-1 more weakly (14, 15), a property described as superselectivity (16).

In this present study, we have investigated how the HA glycocalyx is retained and organized on the surface of DCs, and how this contributes to the recently established role of the glycocalyx in regulating entry and trafficking in lymphatic vessels. We reveal that the glycocalyx is tethered exclusively by the key leukocyte receptor CD44, which configures the constituent HA polymers for efficient docking and adhesion via LYVE-1$^+$ transmigratory cups. Furthermore, using a mouse model of skin contact hypersensitivity, we show that $CD44$ gene deletion in DCs impedes their trafficking to draining LNs by disrupting their capacity to adhere and enter initial lymphatic capillaries. Finally, by inducing an increase in the HA-binding capacity of CD44 using potentiating antibody in vivo, we show the resulting increase in HA glycocalyx density dramatically enhances DC docking to LYVE-1 and promotes their arrest within the vessel lumen. These results identify a previously unrecognized role for CD44 in DCs and reveal its importance in the interplay between HA and LYVE-1 for lymphatic trafficking. In addition, they uncover

---

[1]Medical Research Council (MRC) Human Immunology Unit, MRC Weatherall Institute of Molecular Medicine, John Radcliffe Hospital, University of Oxford, Oxford, UK [2]Wolfson Imaging Centre Oxford, MRC Weatherall Institute of Molecular Medicine, John Radcliffe Hospital, University of Oxford, Oxford, UK

Correspondence: louise.johnson@imm.ox.ac.uk; david.jackson@imm.ox.ac.uk

 

the dynamic nature of the DC HA glycocalyx and reveal a potential mechanism by which DCs might regulate their own trafficking through control of CD44:HA-binding efficiency.

# Results

### CD44 is the primary anchoring receptor for the DC HA glycocalyx

Numerous different non-hematopoietic cell types, including fibroblasts, chondrocytes, vascular smooth muscle, and endothelial cells are surrounded by HA-rich pericellular coats (17, 18), whose anchorage has been attributed to the widely expressed HA receptor CD44 (19) as well as retention by the plasma membrane-bound HA synthases (13, 20). To determine if similar modes of anchorage are used by DCs, we compared the integrity of the HA coat in mouse BMDCs from wild-type and $CD44^{-/-}$ littermates by flow cytometry, using the biotinylated Versican G1 high-affinity HA-binding domain (bVG1) as a probe. We detected abundant HA on the surface of $CD44^{+/+}$ BMDCs, which increased further (up to twofold) upon LPS-induced maturation (as shown previously (12), and Fig 1A and B). In contrast, BMDCs from $CD44^{-/-}$ littermates had typically fourfold lower levels of HA at the cell surface, and these remained unchanged upon maturation (Fig 1A and B). However, $CD44^{-/-}$ DCs displayed almost identical surface levels of MHC class II and the key co-stimulatory receptors CD80 and CD86 to those of their $CD44^{+/+}$ litter- and cage-mates (Fig S1A and B), indicating that gene deletion had no obvious deleterious effects on normal DC immune responsiveness.

Next, we visualized the fine structure of the HA glycocalyx in bVG1-stained BMDCs using high resolution Airyscan confocal microscopy. As shown in Fig 1C, this was visible on the surface of wild-type BMDCs as a dense pericellular coat enmeshed with and extending to a median thickness of 450 nm (Fig 1E), whereas no such HA structure was apparent in $CD44^{-/-}$ BMDCs (Fig 1D and E). Importantly, bVG1 yielded highly specific staining for HA, which could be removed by prior treatment with hyaluronidase (HAase) (Fig S2A and B).

To affirm equal involvement of CD44 in assembly of an HA glycocalyx in native MHC class II$^+$ CD11c$^+$ DCs, we performed bVG1 staining of cells both in situ in normal mouse dermis, and after ex vivo crawl-out, to generate an activated lymph migratory dermal DC population. Whereas an HA coat was present on almost 60% of such dermal DCs in CD44$^{+/+}$ mice, no HA was detected on the majority (>97%) of $CD44^{-/-}$ dermal DCs (Figs 1F and G and S3A). As expected, HA was also present in abundant amounts within the interstitium. Interestingly, however, the levels were reduced by more than twofold in $CD44^{-/-}$ mice, verifying that CD44 plays a role in retaining HA within the extracellular matrix, most likely through anchorage by stromal cells (Fig S3B and C).

In addition to anchoring the HA glycocalyx in DCs, we also assessed the possibility that CD44 regulates the synthesis and secretion of HA. Western blot analysis showed both immature and mature BMDCs express similarly high levels of the key enzymes HAS1 and HAS2 that synthesize HA chains of high MW (>2,000 kD) (21). However, we detected no HAS3 protein expression, and the levels of all three HAS enzymes were not significantly altered in BMDCs from $CD44^{-/-}$ mice (Figs 2A and S4). We also measured the levels of HA itself, using a sensitive competitive ELISA. This showed the glycosaminoglycan was present in similar amounts in whole cell lysates (cell surface and intracellular fractions) of immature BMDCs from both $CD44^{+/+}$ and $CD44^{-/-}$ and increased almost twofold in each case after LPS-induced maturation (Fig 2B and C). Moreover, Airyscan confocal imaging of permeabilized cells revealed that HA in CD44$^{+/+}$ BMDCs was predominantly at the cell surface, whereas in $CD44^{-/-}$ BMDCs, HA accumulated in intracellular vesicles (Fig 2D), suggestive of a role for CD44 in their trafficking to the plasma membrane. Surprisingly, HA was also secreted in significant amounts by both $CD44^{+/+}$ and $CD44^{-/-}$ BMDCs, with both cell-associated HA and soluble HA showing a similar twofold increase upon LPS-induced maturation (Fig 2B and C). These findings reveal that CD44 is essential for processing, anchorage and retention of an endogenously synthesized HA glycocalyx in DCs. However, CD44 is dispensable for both HA synthesis and secretion.

### CD44 is translocated to the DC uropod for HA-mediated adhesion to lymphatic endothelium

Next, we investigated the functional role of CD44 and its organization of the HA glycocalyx during DC adhesion to lymphatic endothelium. We carried out spinning disc video microscopy to observe initial encounters of CMFDA-labeled human monocyte–derived DCs (MDDCs) with a human lymphatic endothelial cell (hLEC) monolayer, using the non-blocking mAb 6A to visualize LYVE-1. As shown in Videos 1 and 2, MDDCs appeared to first adhere to a discrete cluster of LYVE-1 in the underlying endothelium via one pole of the cell, before extending lamellipodia-like protrusions to probe the surrounding monolayer. Significantly, both attachment to LYVE-1 and protrusion formation were disrupted when MDDCs were pretreated with the CD44 mAb IM7 that induces CD44 shedding (22, 23) or the CD44 HA–blocking mAb BRIC235 (24), both of which destabilize the glycocalyx (Figs 3A and S5A and B and Videos 3 and 4). Furthermore, dual staining of adhering MDDCs for ICAM3, a well-documented marker for the uropod, revealed that CD44 was preferentially localized to this distinct region at the trailing edge of the cell (25) where it formed the primary adhesive interface with underlying LECs (Fig 3B and Videos 5 and 6). We also observed that disrupting CD44–HA interactions at the surface of MDDCs by prior treatment with either IM7 or BRIC235 mAbs abolished the capacity of these cells to establish formation of the characteristic LYVE-1$^+$ endothelial cups that mediate DC-lymphatic adhesion and transmigration (Fig S5C). Hence, the directed distribution of HA glycocalyx to the uropod via CD44 is pivotal to both these critical processes.

To explore the influence of CD44-dependent endothelial adhesion on DC motility, we tracked the migratory paths of MDDCs on hLEC monolayers. In the presence of control IgG, MDDCs crawled briefly but adhered rapidly, inducing LYVE-1 clustering in the regions of cell–cell contact before transmigrating across the monolayer, below the plane of view (Fig 3C and D and Videos 1 and 2). In contrast, MDDCs that were treated with either IM7 or BRIC235 failed to form long-lived interactions and thus crawled further (12-fold and 9-fold, respectively), with longer migratory paths (Fig 3C and D and Videos 3 and 4). Importantly, no effect was observed with the control non-blocking CD44 mAb F.10.44.2 (Fig 3E).

Similarly, we used spinning disc microscopy to simultaneously track the migratory paths of CD44$^{+/+}$ and $CD44^{-/-}$ BMDCs on monolayers of primary dermal mouse lymphatic endothelial cells (mLECs), using cells that had been differentially tagged with either

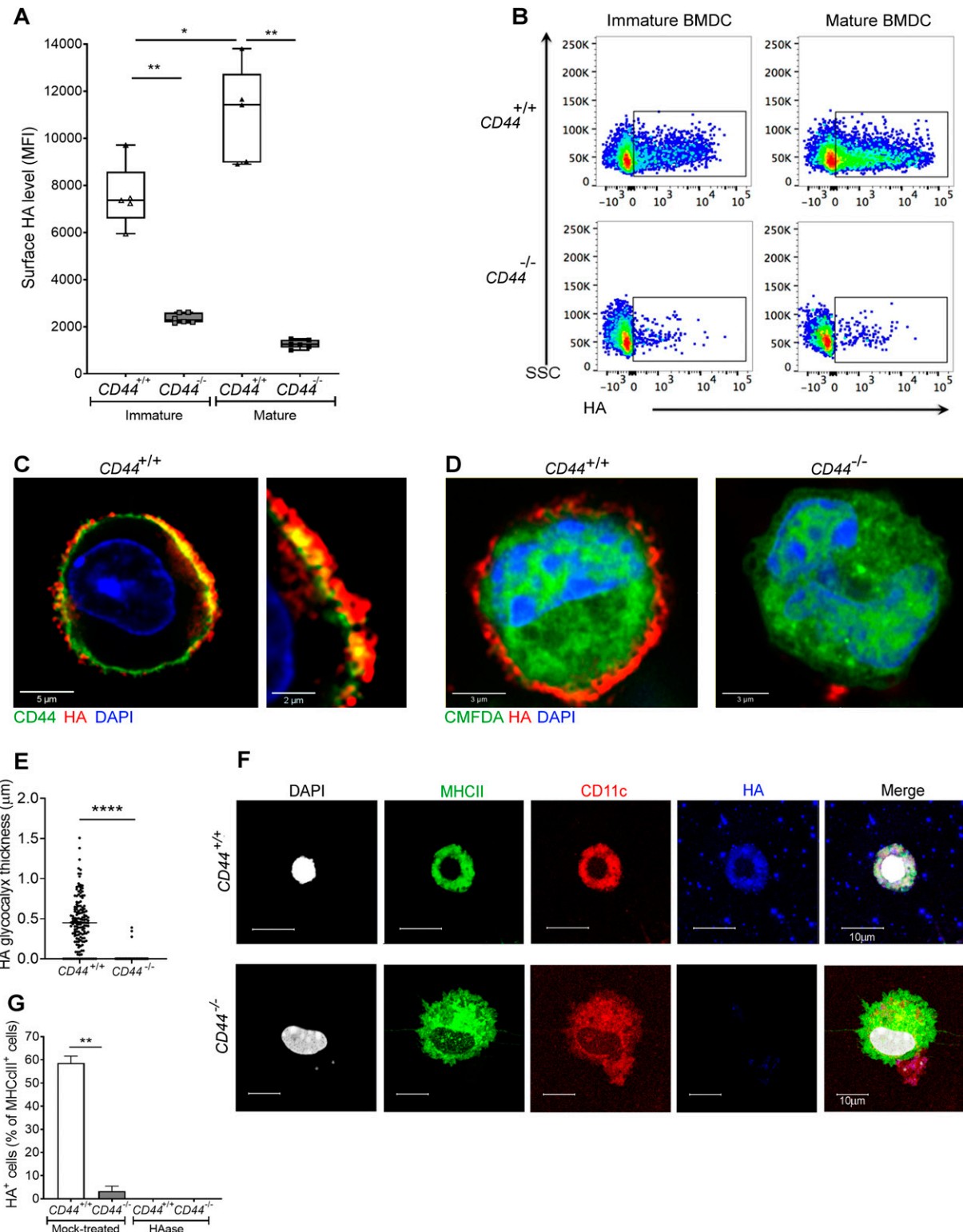

**Figure 1.   CD44 anchors the HA glycocalyx to the surface of both immature and mature DCs.**
**(A)** HA on intact BMDCs prepared from *CD44*[+/+] and *CD44*[−/−] littermates, as detected using bVG1 with streptavidin-AF647 by flow cytometry. Data in box plots represent the median (center bar) ± s.e.m. (whiskers), n = 5, 6 mice. Data are one representative experiment from three. **(B)** Representative dot plots of bVG1-streptavidin-AF647, to indicate levels of cell surface HA. **(C)** Visualization of the HA glycocalyx by bVG1–streptavidin–AF647 (red) on an individual PFA-fixed *CD44*[+/+] BMDC, with CD44 detected by mAb IM7 (green) and nucleus counterstained with DAPI (blue), as observed by confocal microscopy with Airyscan detection, with digital zoom (right panel) to show dimensions of the extended pericellular coat. **(D)** Confocal microscopy of fixed *CD44*[+/+] and *CD44*[−/−] BMDCs labeled with CMFDA green tracker dye, bVG1-streptavidin-

CMFDA green or CMTPX red cell tracker dyes (or vice versa). Wild-type $CD44^{+/+}$ BMDCs adhering to mLECs displayed a highly restricted pattern of migration, which in most cases led to transmigration of the monolayer, whereas $CD44^{-/-}$ BMDCs were only loosely attached and displayed longer migratory paths (Fig 4A and B and Video 7). High magnification confocal microscopy revealed co-localization of CD44 and HA on the BMDC surface, aligning with LYVE-1 within the distinctive ring-like transmigratory cups through which they adhered to underlying mLECs (Fig 5A and B). Although some LYVE1$^+$ ring-like structures were also visible around $CD44^{-/-}$ BMDC, these were more diffuse and fourfold less abundant, with similar numbers to those seen in control, HAase-treated BMDCs (Fig 5B and C). These data show that engagement with the CD44–HA glycocalyx complex is vital for the capture of DCs by lymphatic endothelium and the initiation of diapedesis.

### CD44 is critical for DC entry to dermal lymphatics

To assess the significance of CD44-mediated glycocalyx organization for lymphatic trafficking in vivo, we topically applied FITC and the skin sensitizing agent oxazolone to the shaved abdomens of $CD44^{+/+}$ and $CD44^{-/-}$ mice, then 24 h later harvested draining inguinal and axillary LNs for analysis by flow cytometry (Figs 6 and S6). We recorded a twofold reduction in recovery of FITC$^+$ CD11c$^+$ DCs in $CD44^{-/-}$ mice, in comparison to $CD44^{+/+}$ litter- and cage-mates (Fig 6A–D), both in terms of absolute numbers (Fig 6A) and as their percentage of total cells in draining LNs (Fig S6A–C). The same impact of CD44 gene deletion was also evident in the FITC-bearing CD11c$^+$ MHC class II$^+$ CD103$^+$ migratory DC subsets (Fig 6B) and was accompanied by a significant ($P = 0.037$) reduction in the total cellularity of $CD44^{-/-}$ LNs (Fig 6C). Nevertheless, because CD44 is also expressed by fibroblasts and epithelial cells in mouse skin, we could not exclude the possibility that such effects may have resulted from gene deletion in these other cells types. Therefore, to avoid such ambiguities, we carried out additional lymphatic trafficking experiments in which we labeled $CD44^{+/+}$ and $CD44^{-/-}$ BMDCs with either Q-dot 655 or Q-dot 585 (or vice versa), and injected 1:1 mixtures into the skin of recipient $CD44^{+/+}$ C57BL/6 mice that had been subjected to oxazolone contact hypersensitivity. We then compared their recoveries in draining cervical LNs by flow cytometry after 24 h. The results (Fig 7A) showed an almost twofold reduction in $CD44^{-/-}$ BMDCs in these LNs, both in terms of cell numbers and as a percentage of total LN cellularity (Fig S7A and B). As this reduction in nodal trafficking was comparable with that observed for BMDCs in LYVE-1-deficient mice (12), we reasoned that CD44, like LYVE-1, might contribute to the same initial step in vessel entry. To explore this further, we prepared fixed whole-mount skin sections from mice injected with CMFDA-labeled $CD44^{-/-}$ and $CD44^{+/+}$ BMDCs, and used confocal imaging to compare their distribution in and around lymphatic capillaries close to the injection site. As expected, significant numbers of CMFDA$^+$ BMDCs of both genotypes were still visible 24 h after transfer (Fig 7B). However, unlike $CD44^{+/+}$ BMDCs, where the majority (80%) of vessel-

associated cells were found within the capillary lumen, most $CD44^{-/-}$ BMDCs (60%) had accumulated at the basolateral surface and few if any were visible in the lumen (Fig 7B and C). Likewise, this correlated with a reduction in the number of CMFDA$^+$ $CD44^{-/-}$ BMDCs recovered from draining LNs, in comparison with $CD44^{+/+}$ BMDCs (Fig 7D), as well as a reduction in overall LN cellularity (Fig 7E).

To further consolidate these findings, we performed in vitro adhesion and transmigration assays (12, 26) using immature and LPS-matured BMDCs, and monolayers of primary dermal mLECs. As shown in Fig 8A–D, the numbers of both resting and LPS-matured $CD44^{-/-}$ BMDC adhering to the LEC monolayers after 3 h were, respectively, between two and fourfold lower than equivalent control $CD44^{+/+}$ cells. Importantly, incubation with HAase reduced the adhesion of $CD44^{+/+}$ BMDCs to the same level as that of $CD44^{-/-}$ BMDCs (Fig 8B), confirming that CD44:HA interactions are responsible for supporting such DC-LEC adherence, rather than other ligands of CD44 such as chondroitin sulfate (27, 28), E-selectin (29) macrophage mannose receptor (MR) (30), or osteopontin (31). In addition, the adhesion of wild-type BMDCs was reduced by more than 40% in the presence of excess free HA (Fig 8C). Furthermore, in Transwell assays with mLEC monolayers, both the rate and extent of basolateral-to-luminal transmigration of $CD44^{-/-}$ BMDC were reduced (up to 85%) by comparison with $CD44^{+/+}$ DCs (Fig 8E). However, mAb-induced blockade of LYVE-1:HA interactions did not impair $CD44^{-/-}$ BMDC transmigration, unlike that of $CD44^{+/+}$ BMDC, confirming that in the absence of CD44, diapedesis does not involve LYVE-1:HA interactions (Fig 8F).

Last, we considered the possibility that genetic deletion of CD44 in BMDCs might contribute to defective nodal trafficking by impeding interactions with HA in the dermal matrix, which could support interstitial migration before vessel entry. To address the issue, we modeled the process over 5 h in vitro, using an assay that measured transit of fluorescent (CMFDA-labeled), LPS-matured BMDCs through monolayers of mouse dermal fibroblasts plated on the upper surface of Transwell inserts. However, rather than blocking DC migration, the results (Fig 8G), indicated CD44 gene deletion marginally enhanced the process in comparison to $CD44^{+/+}$ cells, although the difference did not reach statistical significance.

These results demonstrate that CD44 retains and organizes the DC HA glycocalyx, to permit efficient docking with lymphatic endothelium and transit to the vessel lumen.

### Increasing HA glycocalyx density through mAb-induced CD44 clustering enhances DC-lymphatic adhesion and leads to arrest in the vessel lumen

The importance of the CD44:HA glycocalyx complex for docking and entry of DCs to dermal lymphatic vessels raised the possibility that functional regulation of the receptor might exert control over lymphatic trafficking in vivo. Indeed, in other leukocyte populations, it is well documented that the HA-binding capacity of CD44 is

---

AF647 (red), and counterstained with DAPI (blue). **(E)** Thickness of the HA glycocalyx on fixed $CD44^{+/+}$ and $CD44^{-/-}$ BMDCs, as measured from Airyscan images using Image J (n = 5, 5 cells, data combined from three experiments). **(F)** Detection of HA (blue) on migratory MHC class II$^+$ (green) CD11c$^+$ (red) dermal DCs, with nuclei counterstained with DAPI (gray), following egress from tissue during ex vivo culture. **(G)** Percentage of MHC class II$^+$ dermal DCs scored as surface HA$^+$ by microscopy (n = 6, 6 mice, data combined from two experiments), following fixation of freshly resected ear skin in PFA and incubation for 2 h at 37 °C ± hyaluronidase (HAase), before immunostaining *$P < 0.05$, **$P < 0.01$, ****$P < 0.0001$, Mann–Whitney $U$-test.

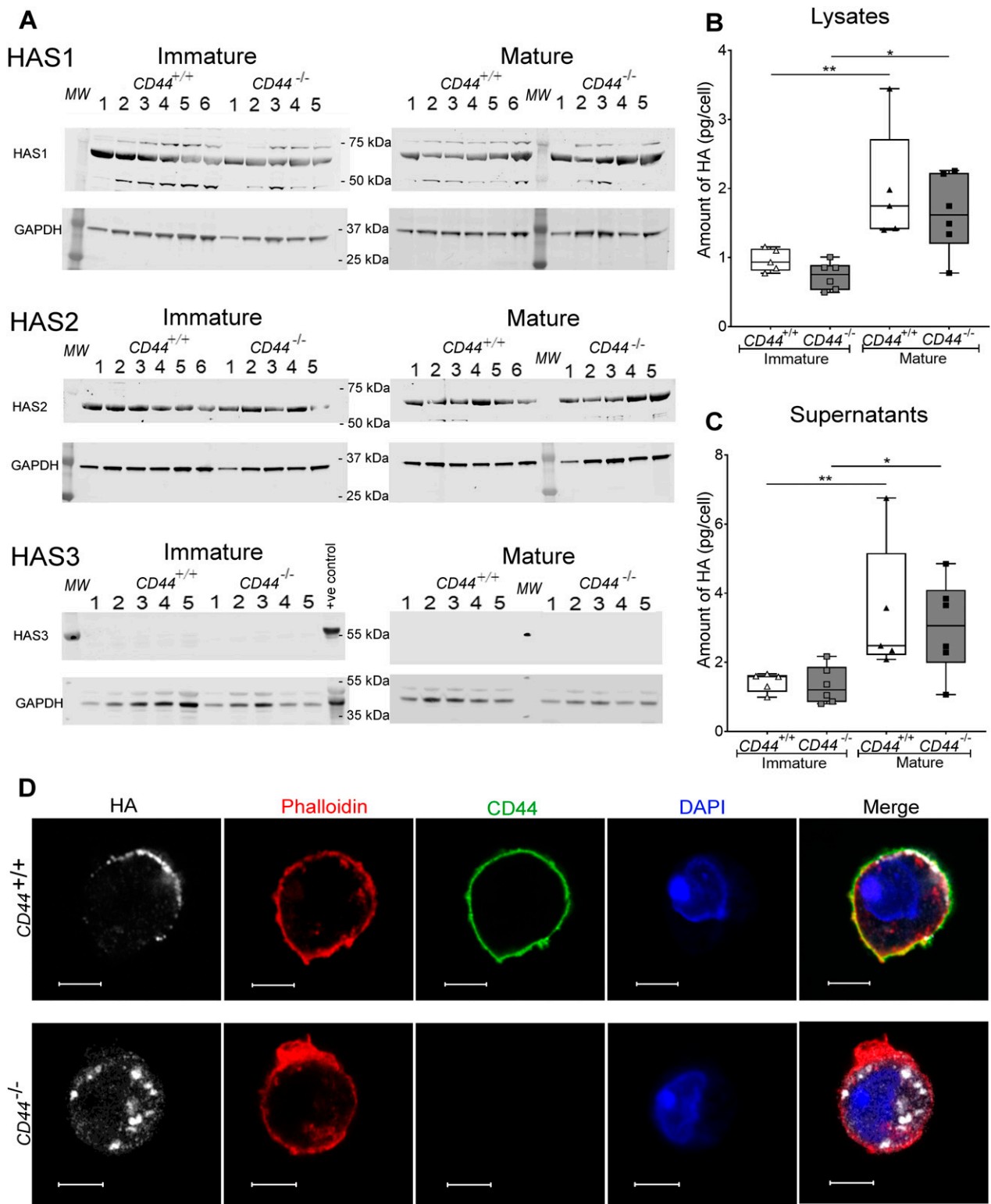

**Figure 2. CD44$^{-/-}$ BMDCs exhibit no defect in HA synthesis.**
**(A)** Cell lysates of immature and mature BMDCs from either five or six individual CD44$^{+/+}$ and five CD44$^{-/-}$ littermates were resolved by SDS–PAGE and probed by Western blotting to detect HAS enzymes HAS1-3 and GAPDH (as a loading control). Representative blots are shown, from one of two separate experiments. Lysate from wild-type mouse skin was included as a positive control for HAS3. **(B, C)** Quantitation of HA in lysates (B) and supernatants (C) of immature and LPS-matured BMDCs, as determined by ELISA. Data represent the median (center bar) ± s.e.m. (whiskers), n = 5 for CD44$^{+/+}$ and 6 for CD44$^{-/-}$ mice, one representative experiment from three separate experiments. **(D)** Detection of HA in permeabilized BMDCs prepared from CD44$^{+/+}$ and CD44$^{-/-}$ littermates, using bVG1 and streptavidin–AF647 (gray), immunostaining for CD44 (green), and counterstaining with phalloidin (red) and DAPI (blue). Scale bars = 5 μm.

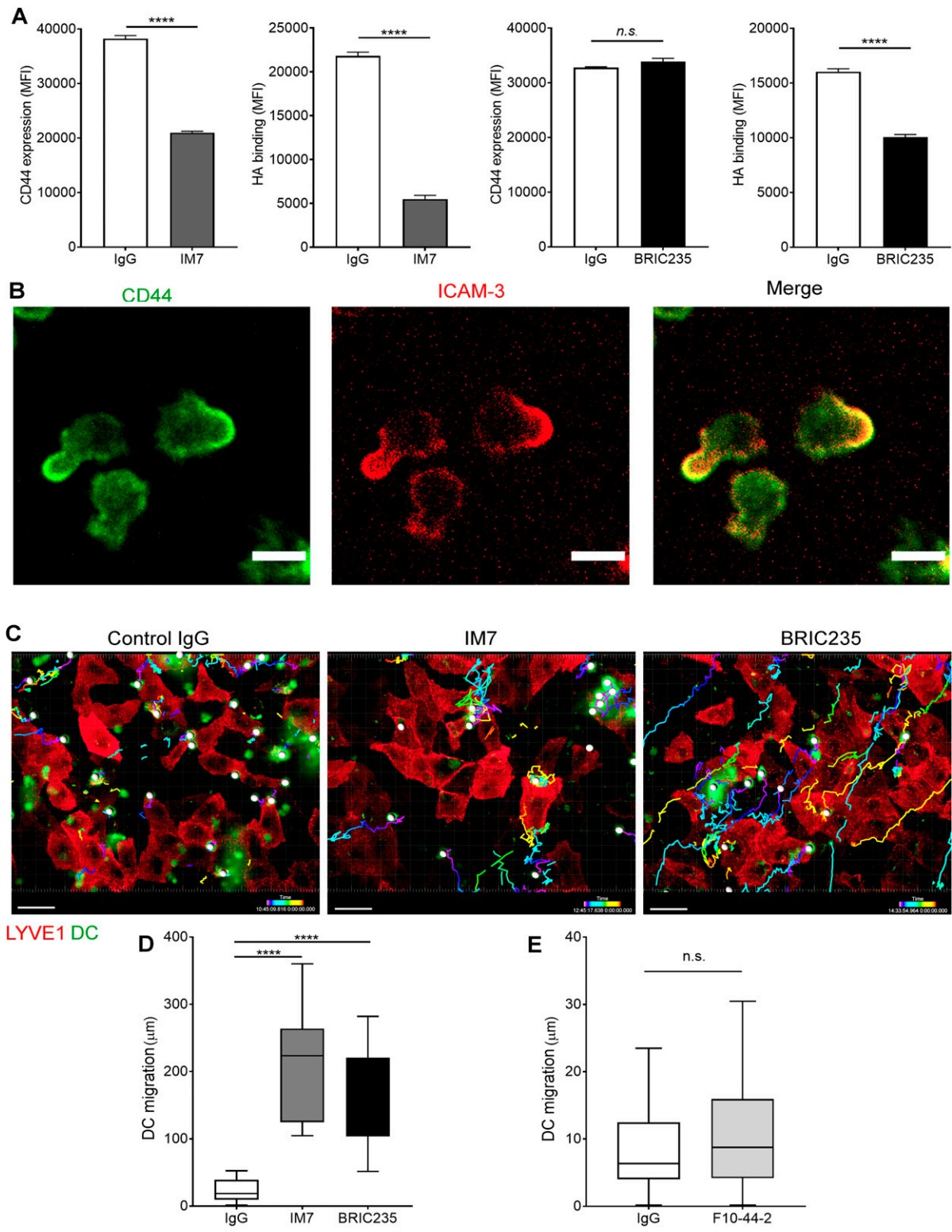

**Figure 3. Disrupting CD44-HA binding impairs DC interactions with human lymphatic endothelial cells (hLECs).**
**(A)** Surface levels of CD44 and HA on LPS-matured human monocyte-derived DC (MDDC) after incubation with control IgG, or anti-CD44 mAbs IM7 and BRIC235, as measured by flow cytometry. ****$P < 0.0001$, unpaired $t$ test. Data are the mean ± s.e.m. (n = 3), one representative experiment of three. **(B)** Single time point from a video of MDDCs labeled with FITC-conjugated anti-CD44 (mAb F10.44.2, green) and APC-conjugated anti–ICAM3 (red) crawling on unstained hLEC monolayers. Scale bar = 10 $\mu$m. **(C, D, E)** Migration of mAb-treated MDDCs on monolayers of hLECs, imaged by spinning disc confocal microscopy over 4 h, with dragon tail tracks color-coded to indicate time progression (C) and quantitated using Imaris software (D, E). Scale bar = 20 $\mu$m, ****$P < 0.0001$, Mann–Whitney $U$-test. Data are the median (center bar) ± s.e.m. (whiskers), data combined from three experiments.

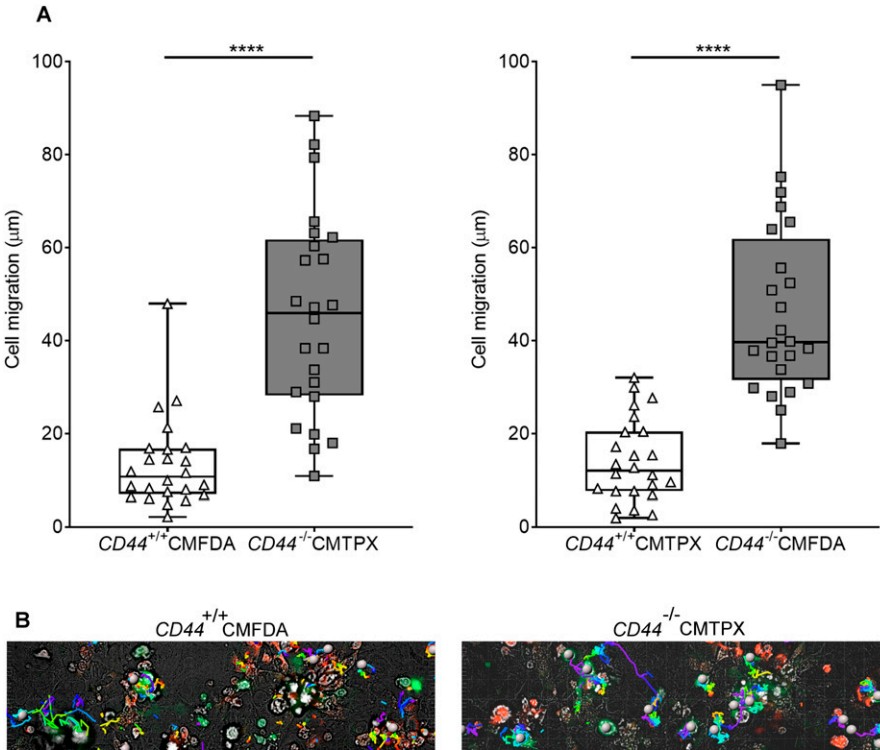

**Figure 4. CD44 deficiency impairs BMDC interactions with mouse lymphatic endothelial cells.**
Tracking migration of LPS-matured BMDCs from $CD44^{+/+}$ and $CD44^{-/-}$ mice, labeled with CMFDA green or CMTPX red tracker dyes, applied to monolayers of mouse lymphatic endothelial cells and imaged by spinning disc confocal microscopy over 4 h. **(A)** Lengths of migratory paths were tracked using Imaris software. ****$P <$ 0.0001, Mann–Whitney $U$-test. Data are the median (center bar) ± s.e.m. (whiskers) eight cells/field of view, three fields of view, data combined from three experiments. **(B)** Representative images of dragon tail paths to indicate time are shown for $CD44^{+/+}$ BMDCs (CMFDA, green) and $CD44^{-/-}$ BMDCs (CMPXT, red) with bright-field, and cells indicated by gray spheres. Scale bar = 50 $\mu m$.

subject to regulation by pro-inflammatory cytokines, through a combination of inside-out signaling, post-translational modification and surface clustering (32, 33, 34). To explore the possibility in more detail, we assessed the consequences of increasing the HA-binding capacity of CD44 in mouse BMDCs using the anti-CD44 mAb IRAWB14, which has been shown to exert such effects in T lymphoma cells by promoting optimal CD44 clustering via an epitope in the N-terminal Link domain (35, 36, 37). As shown in Fig 9A, incubation of wild-type $CD44^{+/+}$ BMDCs with IRAWB14 led to an almost twofold increase in their capacity to bind exogenously added HA. More compellingly, IRAWB14 also increased the extent to which BMDCs bound endogenously synthesized HA, as evidenced by an almost sixfold increase in surface HA levels in the absence of any externally added HA (Fig 9B). This latter finding indicates that the density of the HA glycocalyx can be altered by changes in CD44 functional status without any change in the levels of HA synthesis. Indeed, we found that most HA synthesized by LPS-matured $CD44^{+/+}$ DCs is secreted (median 2.4 pg/cell), and less than 50% (median 1.7 pg/cell) is retained by CD44 tethering (Fig 2B and C). In contrast, we detected no capacity for binding exogenous HA in $CD44^{-/-}$ BMDCs, nor any potentiation of endogenous or exogenous HA binding in response to IRAWB14, confirming that the effects of this mAb are CD44 dependent (Fig 9A and B).

Next, to determine the functional consequences of increased HA glycocalyx density, we subjected both control and IRAWB14-treated BMDCs to in vitro adhesion and transendothelial migration assays, whereby CMFDA green tracker dye-labeled LPS-matured BMDCs from wild-type mice were applied to monolayers of primary mLEC plated in gelatin-coated multiwell dishes and on the undersurface of opaque Transwell inserts, respectively. Significantly, treatment (3 h) with IRAWB14 increased adhesion of BMDCs almost fourfold in comparison with controls, as assessed by quantitative fluorescence measurement (Fig 9C) and confocal microscopy (Fig 9D), through LYVE-1+ transmigratory cup formation (Fig 9E and F). Notably, however, IRAWB14-treated BMDCs displayed a slower rate of basolateral-to-luminal migration across mLEC monolayers, with a significant (up to twofold) reduction in numbers of transmigrating cells (Fig 9G). These results establish that increasing HA glycocalyx density through induced CD44 clustering has a marked effect on DC behavior, altering DC-endothelial adhesion and transmigration in a reciprocal manner.

Finally, we explored the consequences of inducing an increase in HA glycocalyx density on DC trafficking in vivo. Accordingly, we treated CMFDA-labeled LPS-activated wild-type BMDCs with either control rat IgG or IRAWB14 for 30 min and injected them separately into the dermis of oxazolone-treated mice. Importantly, such

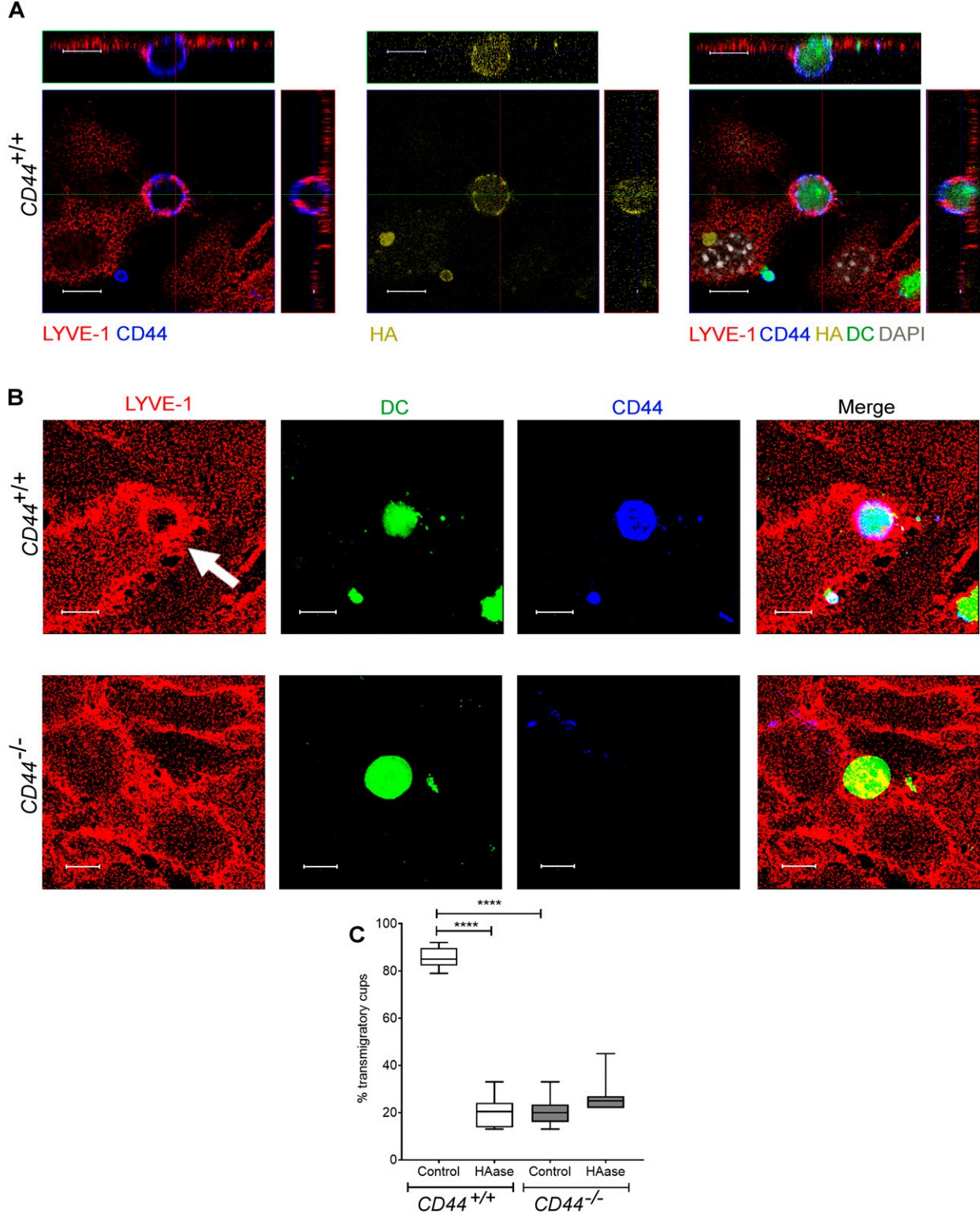

**Figure 5. CD44 is required for efficient formation of lymphatic endothelial transmigratory cups.**
**(A)** Confocal microscopy orthogonal view images of cultured primary mouse lymphatic endothelial cell (mLEC) monolayers immunostained with anti–LYVE-1 (red) viewed 3 h after the addition of LPS-matured, fluorescently labeled (green) *CD44*$^{+/+}$ BMDC immunostained with anti-CD44 (blue), and bVG1 (yellow) and counterstained with DAPI (gray), scale bar = 10 $\mu$m. **(B)** Three-dimensional rendering of confocal images of cultured primary mLEC monolayer immunostained with anti–LYVE-1 (red) viewed 3 h after the addition of fluorescently labeled (green) *CD44*$^{+/+}$ or *CD44*$^{-/-}$ BMDC immunostained with anti-CD44 (blue). An individual LYVE-1-lined transmigratory cup is indicated by arrow. Scale bar = 10 $\mu$m. **(C)** Quantitation of microscopy images to show the number of LYVE-1$^+$ transmigratory cups associating with adherent DCs, after 2-h preincubation with HAase and 3-h co-culture of mLEC monolayers with either mature *CD44*$^{+/+}$ or *CD44*$^{-/-}$ BMDC. ****$P < 0.0001$, Mann–Whitney *U*-test. Data are the median (center bar) ± s.e.m. (whiskers) (n = 10 fields of view), data combined from three experiments.

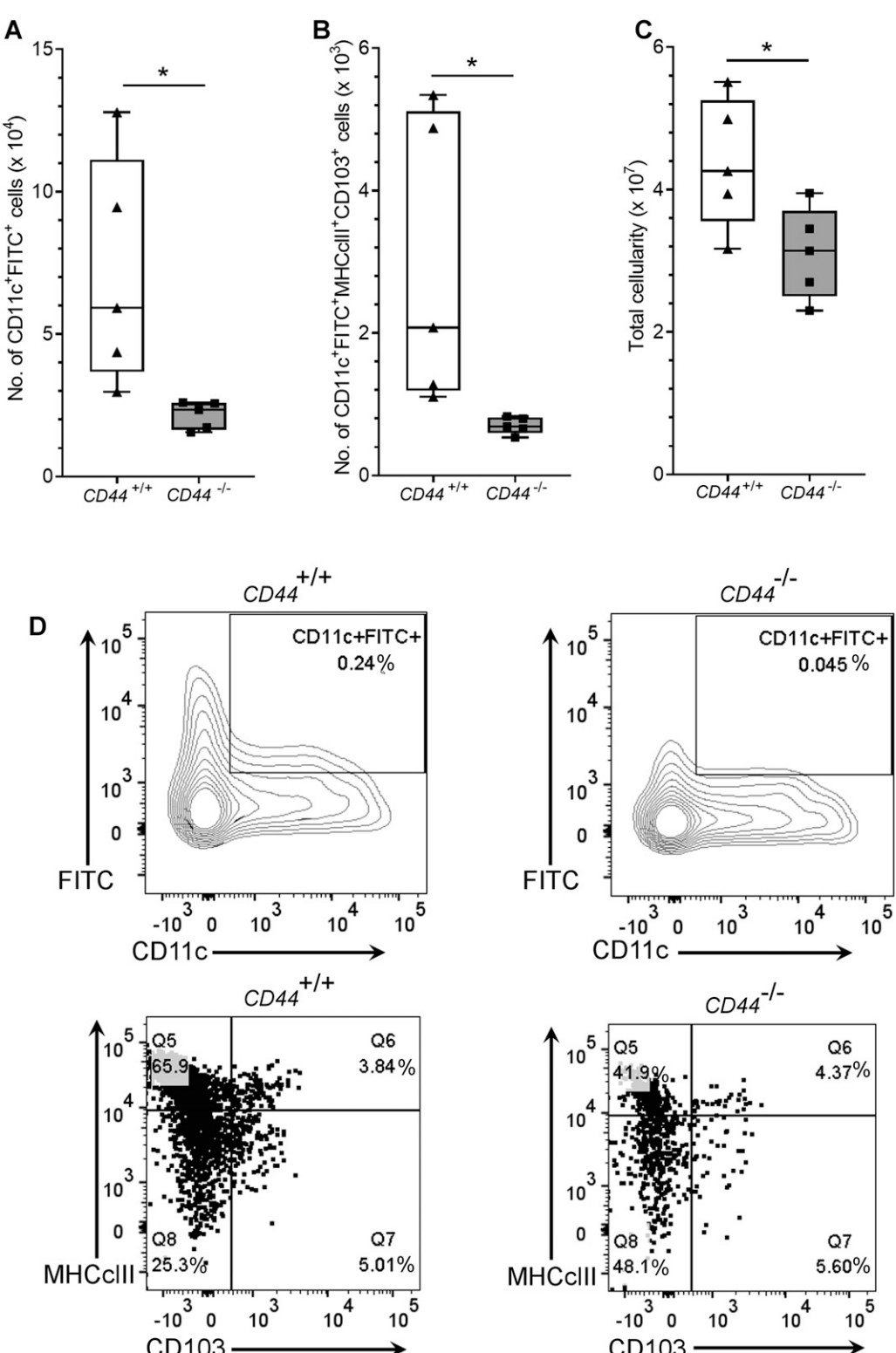

**Figure 6.   Impaired trafficking of endogenous dermal DCs to draining LNs in CD44-deficient mice.**
**(A, B)** Recovery of endogenous DCs from draining inguinal and axillary LNs, 24 h after topical application of oxazolone and FITC, as measured by flow cytometry. **(A, B)** Live CD45⁺ CD11c⁺ FITC⁺ cells (A) were further gated according to expression of MHC class II and CD103 (B). **(C)** Total cellularity of LNs was measured by counting cells with a hemocytometer. *$P < 0.05$, Mann–Whitney $U$-test. Data represent the mean (center bar) ± s.e.m. (whiskers) (n = 5 mice), one representative experiment of three. **(D)** Representative contour and dot plots, showing percentage of cells in each gate.

incubation with IRAWB14 did not affect cell viability or induce CD44 shedding, as the levels of bound antibody remained undiminished after 24 h (Fig S8A). Next, we assessed the ability of IRAWB14-treated BMDCs to enter lymphatic capillaries and migrate to draining cervical LNs by confocal microscopy and flow cytometry, 24 h after injection. The results (Fig 10A and B) revealed a fourfold decrease in IRAWB14-treated BMDCs recovered from skin-draining LNs compared with IgG-treated controls, both in terms of absolute numbers and as percentages of the total LN cell population, consistent with the slower rate of transmigration observed in the in vitro assays (Figs 9G and S8B). This was accompanied by a fourfold decrease in total LN cellularity in mice receiving IRAWB14-treated BMDC, in comparison to those receiving IgG control cells (Fig 10C). Moreover, detailed examination of the dermis by confocal microscopy (Fig 10D–F) showed a threefold greater accumulation of IRAWB14-treated BMDCs inside the lumen of lymphatic capillaries (median 9 BMDCs/100 $\mu$m) compared with IgG-treated controls (median 3 BMDCs/100 $\mu$m), and an enlarged vessel diameter (median 23 $\mu$m diameter, compared to 17 $\mu$m in controls). This is likely due to occlusion of lymph flow by trapped BMDCs, although clearly, we cannot rule out the possibility that the IRAWB14 mAb may also evoke bystander inflammatory effects that affect lymph flow.

These results show that increasing the density of the HA glycocalyx through clustering and functional activation of CD44 can indeed influence the efficiency of DC trafficking via lymph, where the heightened avidity of the CD44:HA complex for LYVE-1 can impede the process by slowing diapedesis and inducing arrest of the migrating cells within the vessel lumen. Furthermore, they establish that CD44 together with HA and LYVE-1 contribute to a critical tripartite adhesion axis that can regulate the rate of DC entry and trafficking via lymph.

## Discussion

Here, we have identified a novel and previously unrecognized role for the leukocyte HA receptor CD44 in regulating exit of DCs from the skin via afferent lymphatics, during their migration to downstream draining LNs. In particular, we have established that CD44 is required for both anchorage and functional organization of a dense HA glycocalyx on the surface of DCs, which enables them to dock and enter lymphatic capillaries. Moreover, we have provided new evidence that CD44 plays an active role in DC trafficking, whereby its ability to modulate glycocalyx density through receptor clustering allows DCs to regulate their adhesion and transmigration of lymphatic endothelium and hence their rate of transit to draining LNs for immune activation. Although previous studies have addressed the role of CD44 in epidermal Langerhans cell trafficking, such investigations used CD44 knockout mice in which tissue-wide deletion of the receptor limited interpretation of the experimental findings (38, 39). In contrast, our present study focused more specifically on comparisons between $CD44^{-/-}$ and $CD44^{+/+}$ DCs and their migration via lymph in recipient wild-type littermates. Such approaches allowed us to conclude that CD44 fulfills a cell-autonomous function in DC trafficking by orchestrating critical interactions between the HA glycocalyx and LYVE-1 in lymphatic vessels.

Widely expressed in different tissues and with multiple roles in cell growth, survival, differentiation and motility, CD44 has a well-documented function in leukocyte extravasation from inflamed blood vessels. Notably, its presence on T-cells and neutrophils mediates their capture from flow and adhesive rolling on HA in the luminal glycocalyx that is itself retained by CD44 expressed in vascular endothelium (19, 24, 40, 41, 42, 43), reviewed in reference 44. However, we have described a different role for CD44 in the exit of DCs via dermal lymphatic vessels in that the HA it engages is not sequestered on lymphatic endothelium but rather synthesized by the DC itself, and its function is to mediate intravasation and not extravasation. Indeed, lymphatic vessels do not possess an HA glycocalyx, neither in vitro nor in vivo (45, 46), nor express CD44. Also, lymphatic endothelium does not use endogenously generated HA for DC adhesion (12). The retention of an HA glycocalyx by CD44 in DCs rather than endothelium and its recognition by a functionally distinct HA receptor LYVE-1 marks a clear difference between lymphatics and blood, and likely evolved to permit independent regulation of leukocyte trafficking in the two distinct vasculatures.

As we showed by Western blotting, synthesis of the HA glycocalyx by BMDCs involves the two glycosyltransferases HAS1 and HAS2, both of which are highly processive (10 sugars/second) and can generate polymer chain lengths of several hundred to several thousand disaccharides, respectively (13, 21, 47, 48). We found that the levels of HA synthesis increased some twofold upon LPS-induced activation of BMDCs, in keeping with the enhanced lymphatic migratory capacity exhibited by mature DC populations in vivo (49). Curiously, however, this increase was not triggered by a rise in HAS enzyme levels and hence may result from an increase in HAS catalytic activity through post-translational modifications such as phosphorylation (50, 51), O-GlcNAcylation or a rise in UDP-sugar substrate levels through increased glucose uptake and cellular metabolism (20, 52, 53).

As revealed by the confocal imaging studies in this present article, the dense CD44-anchored HA glycocalyx likely extends over 500 nm from the DC surface, consistent with the dimensions of CD44-bound HA films in a supported lipid bilayer system (54). This is well beyond the sphere of interaction of smaller surface adhesion molecules such as the key $\beta$2 integrin LFA-1 which has an extracellular domain of 20 nm (55, 56), or even selectins (50–100 nm) (57, 58). Hence, we predict that the HA glycocalyx is poised to make the first adhesive contacts between DCs and lymphatic endothelium, and initiates docking via LYVE-1-enriched transmigratory cups (12), reviewed in reference 9. This likelihood is further substantiated by our observations in the current study, that $CD44^{-/-}$ DCs were unable to initiate stable adhesion to lymphatic endothelial monolayers and instead migrated randomly over the endothelial surface. Also, we showed that contacts between LYVE-1 and CD44:HA formed predominantly at the trailing edge of the DC, within the uropod. Extension of this dynamic, foot-like protrusion which is characteristic of motile cells (59) was previously shown to facilitate adherence and transmigration of leukocytes in vascular endothelium (60, 61). Moreover, in T cells and neutrophils, CD44 itself regulates polarization, assembly and stabilization of the uropod during adhesion to extracellular matrix (60, 62). The receptor is translocated to the posterior pole of these cells by its association with phosphorylated Ezrin, Radixin, and Moesin proteins, triggering membrane protrusion through cortical actin assembly/disassembly via

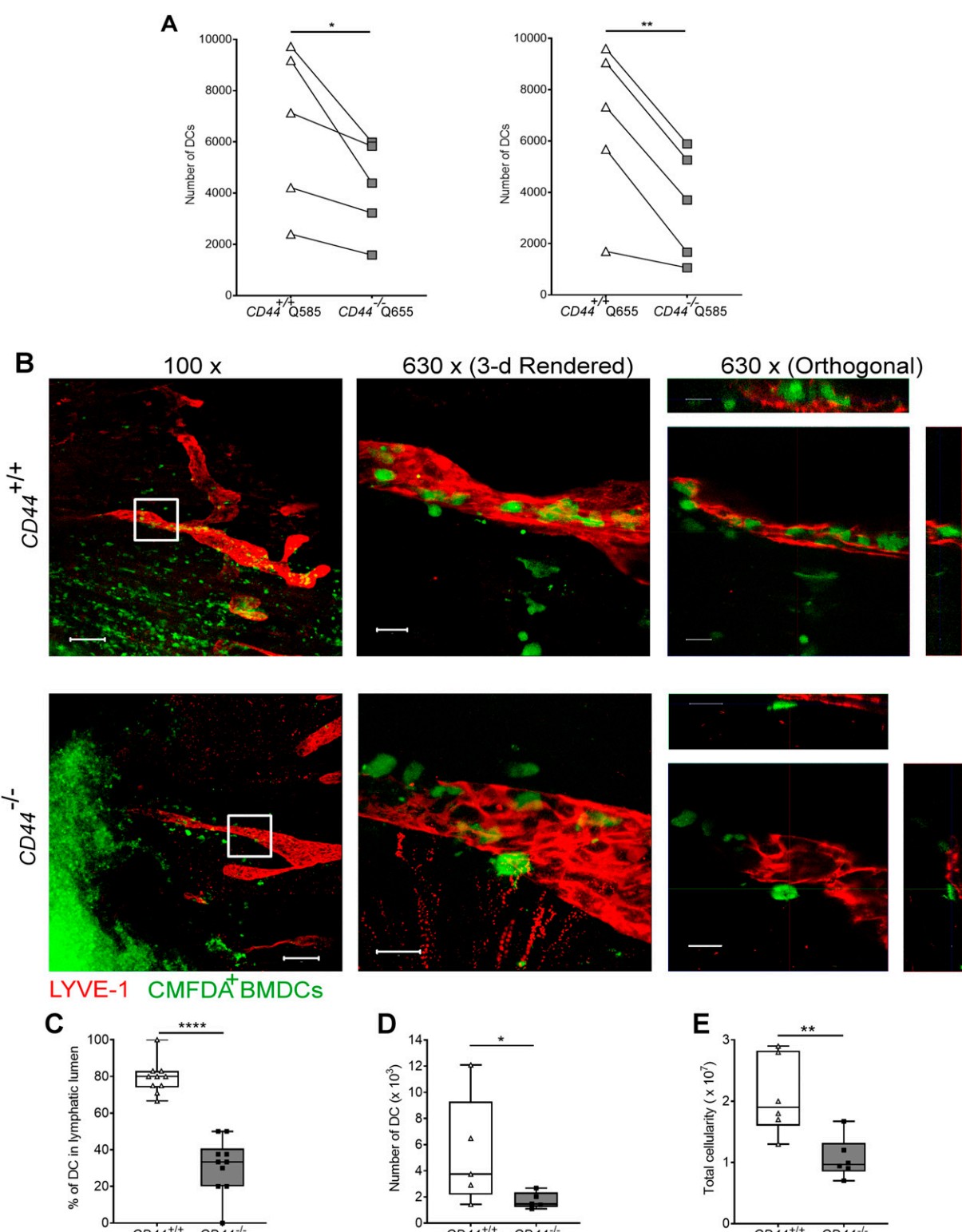

**Figure 7.  Impaired entry and trafficking of CD44$^{-/-}$ BMDCs in dermal lymphatic vessels.**
**(A)** Comparison of lymphatic trafficking of LPS-matured *CD44*$^{+/+}$ and *CD44*$^{-/-}$ BMDCs, differentially labeled with Q-dot 585 and 655, respectively, or vice versa, 24 h after intradermal co-injection into oxazolone-painted skin of *CD44*$^{+/+}$ mice. Recovery of BMDCs in draining cervical LNs was measured by flow cytometry, *$P < 0.05$, **$P < 0.01$, Paired *t* tests. **(B)** Entry of CMFDA-labeled *CD44*$^{+/+}$ and *CD44*$^{-/-}$ BMDCs into dermal afferent lymphatics immunostained with anti–LYVE-1, following topical administration of oxazolone. Panels show 3-D rendering of z-stacks at low magnification (left; 100×, scale bars = 100 $\mu$m), and higher magnification (middle; 630×, scale bars = 20 $\mu$m) with orthogonal sections (right). **(C)** Numbers of BMDCs inside lymphatic vessel lumens, expressed as a percentage of the number of lymphatic vessel-associated BMDCs. Data combined from three experiments, ****$P < 0.0001$, Mann–Whitney *U*-test. **(D, E)** Recovery of intradermally injected CMFDA-labeled BMDCs (D) and overall cellularity (E) in draining cervical LNs 24 h after topical application of oxazolone and adoptive transfer of BMDCs, as measured by flow cytometry. *$P < 0.05$, **$P < 0.01$, Mann–Whitney *U*-test. Data are the median (center bar) ± s.e.m. (whiskers) (n = 5 mice). One representative experiment of three.

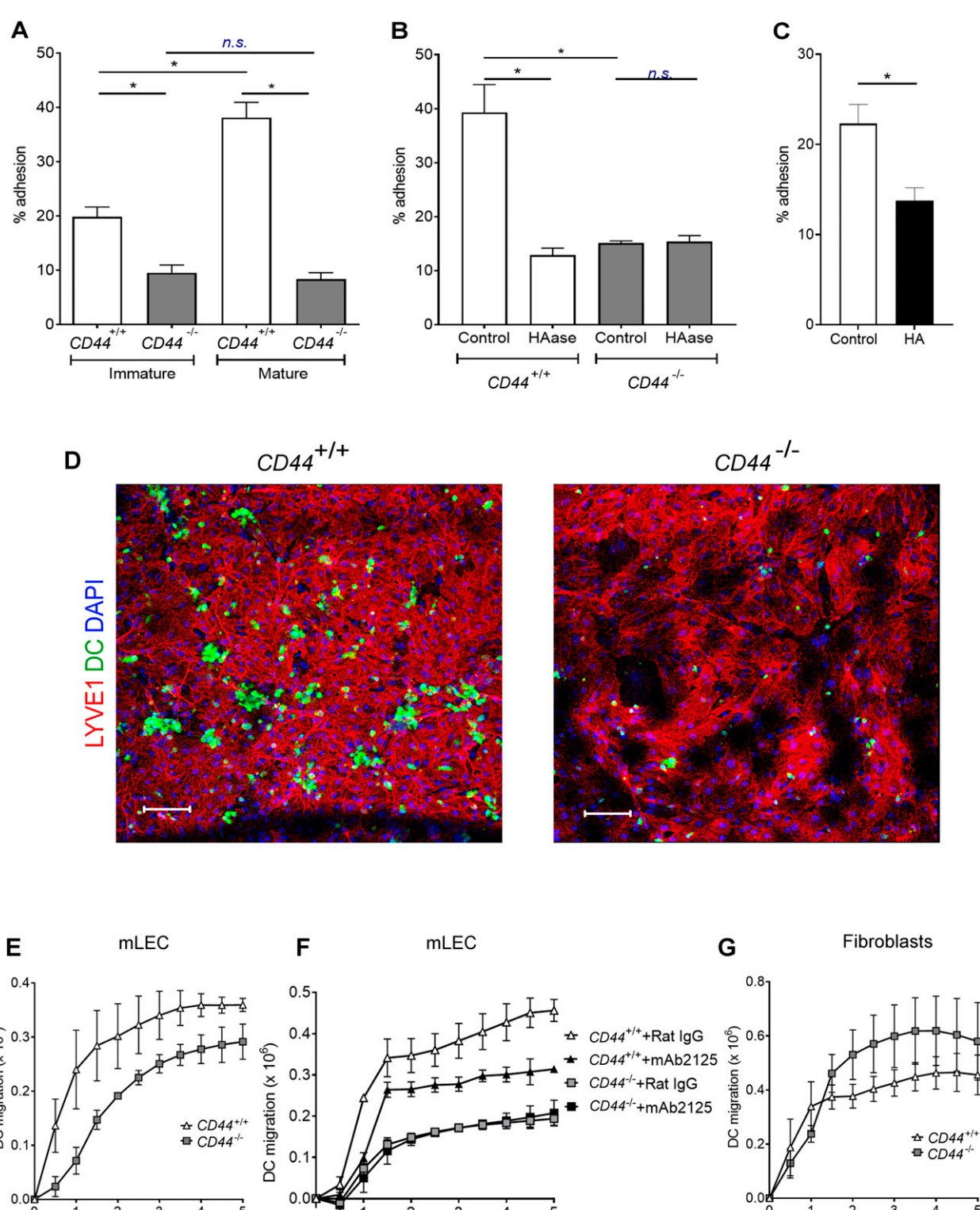

**Figure 8. DC adhesion and transendothelial migration are dependent on CD44 and HA.**
**(A, B, C)** Adhesion of LPS-matured CMFDA-labeled CD44$^{+/+}$ and CD44$^{-/-}$ BMDCs to mouse lymphatic endothelial cell (mLEC) monolayers after 3 h incubation, as assessed by fluorescence plate reader, comparing (A) immature and mature BMDC, (B), mature BMDC after 2-h incubation with or without hyaluronidase (HAase) and (C) mature wild-type BMDC in the presence of exogenously applied high molecular weight HA. **(D)** Immunostaining and confocal microscopy of mLEC monolayers following 3 h co-culture with CD44$^{+/+}$ or CD44$^{-/-}$ LPS-matured CMFDA-labeled BMDC (green), with anti–LYVE-1 (red) and nuclei counterstained with DAPI (blue). Scale bar = 100 $\mu$m. **(E, F, G)** Transmigration of LPS-matured CMFDA-labeled CD44$^{+/+}$ and CD44$^{-/-}$ BMDCs through either mLEC monolayers cultured on the undersurface of Transwell filters in the presence of control rat IgG or anti–LYVE-1 neutralizing mAb 2125 where indicated (E, F) or across fibroblast monolayers cultured on the upper surface of filters (G) measured over a 5-h period by fluorescence plate reader. *$P$ < 0.05, Mann–Whitney $U$-test. Data are the mean (center bar) ± s.e.m. (whiskers) (n = 4), representative experiments of three separate experiments.

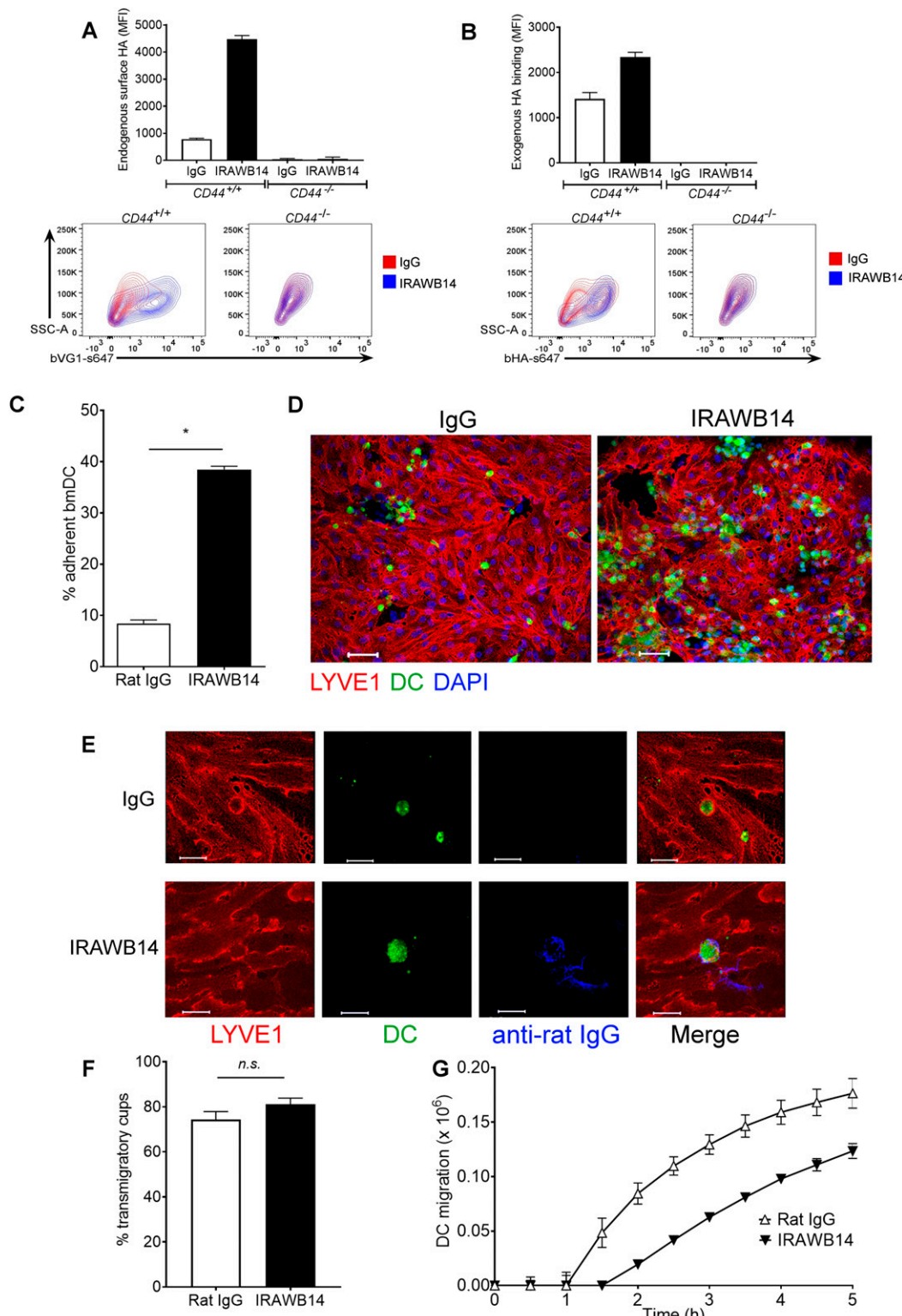

**Figure 9. mAb-induced enhancement of CD44-HA binding increases BMDC adhesion to mouse lymphatic endothelial cell (mLEC) and impairs transmigration.**
**(A, B)** Capacity of LPS-matured $CD44^{+/+}$ and $CD44^{-/-}$ BMDCs to bind endogenously synthesized HA (A) or exogenously added HA (B), assessed by incubation with bVG1 and streptavidin–AF647 or bHA and streptavidin–AF647, respectively, with quantitation by flow cytometry and showing representative contour plots. **(C, D)** Adhesion of LPS-matured CMFDA-labeled wild-type BMDCs (green) to mLEC monolayers after 3-h incubation in the presence of either rat IgG or the CD44 potentiating antibody IRAWB14, as assessed by fluorescence plate reader (C) and confocal microscopy (D), with mLEC immunostained for LYVE-1 (red) and counterstained with DAPI (blue). Scale bar = 50 $\mu$m. **(E, F)** Effect of IRAWB14 or control rat IgG on transmigratory cup formation in mLEC monolayers co-cultured for 3 h with LPS-matured CMFDA-labeled

Rho and its associated kinase ROCK (63, 64). In our present study, the dense distribution of CD44 and its associated HA glycocalyx in the DC uropod likely facilitates efficient capture by LYVE-1, consistent with the known avidity-dependent binding properties of the receptor and its low affinity for free HA polymers (9, 15, 65). In addition, as demonstrated for matrix-adherent tumor cells that can also assemble a dense HA glycocalyx via CD44 (66), the extended dimensions of the bulky CD44:HA complexes on the surface of DCs may exert biophysical influences on the smaller underlying DC integrins by constraining their lateral diffusion, driving their clustering and promoting their activation through tensile stress. Given the key roles of $\beta1$ and $\beta2$ integrins in DC:LEC adhesion and transmigration under conditions of inflammation (26), and the observation that their counter-receptors ICAM-1 and VCAM-1 are present within LYVE-1-rich lymphatic endothelial transmigratory cups (12, 67), it is likely that CD44 may also help choreograph DC transit by coordinating initial HA glycocalyx-mediated contacts with endothelium and subsequent firm integrin-mediated adhesion.

In comparison with afferent lymphatic migration of DCs, much less is understood about the mechanisms governing that of T cells (reviewed in reference 68), despite the fact they comprise 85–90% of cells in afferent lymph (69, 70). As T-cells exhibit lower levels of surface HA than DCs (71), it is unclear whether they have a sufficiently dense HA glycocalyx with which to interact with LYVE-1. Instead, lymphatic endothelial expressed-macrophage mannose receptor may be the dominant interaction partner for lymphocyte CD44 (30, 72).

In addition to retaining the HA glycocalyx and targeting its distribution within the pro-adhesive DC uropod, CD44 likely imposes a spatial organization on the bound HA polymers that shapes their interaction with LYVE-1. The notion that HA configuration is critical for receptor binding is supported by numerous experimental findings. Notably the distinctive pericellular HA cables formed by wrapping of the polymer chains around their binding partner versican and covalent attachment to I$\alpha$I (inter $\alpha$ trypsin inhibitor) heavy chain present in cells exposed to inflammatory stimuli, convert HA from a weak to a strongly adhesive state for CD44 on monocytes (73, 74, 75, 76). Likewise, in the case of LECs, complexing HA with TSG-6 dramatically enhances its binding to LYVE-1 (65). Although the precise configuration of HA in the DC glycocalyx is currently unknown, the relatively short CD44 footprint (6–8 sugars (77, 78)) and the likelihood that only a small proportion of these are occupied by the receptor (based on the abundance of free bVG1 binding sites) predict a tethered HA meshwork comprising multiple loops and free ends of varying lengths. That such a topography has special significance for selective binding of the glycocalyx to LYVE-1 is supported by recent biophysical analyses which revealed an unexpected complementarity between the two receptors insofar as LYVE-1 binds much more slowly than CD44 to HA loops and displays a marked preference for HA chain ends (Bano, Banerji, Jackson and Richter unpublished). Whether and how these features enable the unusual sliding mode by which LYVE-1 engages HA as distinct from the conventional sticking interactions of CD44 (79, 80) is the subject of ongoing research in our laboratory.

Finally, by experimentally manipulating the HA-binding capacity of CD44 in DCs, we showed that the receptor can control glycocalyx surface density and consequently adhesion to LYVE-1, conferring on it the potential to regulate DC entry and trafficking through initial lymphatics. Specifically, treatment with the mAb IRAWB14 that cross-links CD44 and potentiates its HA-binding (35, 37, 81) led to increased incorporation of endogenous HA into the glycocalyx, elevating its surface density by almost sixfold and strengthening its adhesiveness for LYVE-1 to such an extent that it significantly delayed DC transmigration. Hence, DCs appear to carry a reservoir of spare HA, for appropriate on-demand expansion of the glycocalyx. Furthermore, when injected into the skin of oxazolone-sensitized mice, we showed that the increased adhesiveness of the HA glycocalyx in IRAWB14-treated DCs led to their accumulation inside the lumen of dermal lymphatic capillaries, effectively halting their migration and subsequent emergence in draining LNs. Following transmigration, DCs normally crawl along the luminal surface of initial lymphatics, before entering downstream contractile vessels for transport to draining LNs (5, 7, 10, 82, 83, 84, 85). However, as LYVE-1 lines the luminal as well as basolateral surfaces of initial lymphatic capillaries (86), our findings raise the intriguing possibility that such crawling may be regulated by interactions between LYVE-1 and the CD44-anchored DC HA glycocalyx. Indeed, regulated retention of DCs within the lumen of initial lymphatics appears important in light of recent reports that DCs pause during such migration to form long-lived MHC-dependent interactions with antigen-specific T-cells (87). We speculate that similar functional modulation of CD44 may occur to control leukocyte trafficking in vivo, in response to cytokines including TNF$\alpha$, IL-2, IL-1, and IFN-$\gamma$ and chemokines such as CCL4 (MIP-1$\beta$), CCL8 (IL-8), and CCL5 (RANTES), already well-documented to activate HA binding by CD44 in monocytes and T-cells (33, 34, 88, 89, 90, 91).

In conclusion, we have shown that CD44 plays a key role in DC migration by anchoring the HA surface glycocalyx, mediating docking onto lymphatic endothelium for vessel entry via the lymphatic HA receptor LYVE-1 in transmigratory cups. Moreover, we have demonstrated that modulation of HA glycocalyx density through CD44 clustering and activation can regulate the efficiency of DC docking and entry to lymphatics, and may possibly influence the rate of DC trafficking to draining LNs in vivo. These findings reveal the intricate nature of the CD44:HA:LYVE-1 axis and highlight its potential as a therapeutic target within the lymphatics for blocking pathological immune activation.

# Materials and Methods

### Human and animal studies

All studies using human tissue were approved by the Oxford Regional Health Committee (OXREC). Animal studies were performed under appropriate UK Home Office licenses according to established institutional guidelines.

---

BMDCs (green) and immunostained for LYVE-1 (red), with bound IRAWB14 (detected with anti-rat IgG, blue), shown in (E), scale bar = 20 $\mu$m, and the percentage of adherent BMDCs associated with LYVE-1* transmigratory cups in each condition shown in (F). **(G)** Effect of IRAWB14 mAb or control rat IgG on basolateral-to-luminal transmigration of LPS-matured CMFDA-labeled wild-type BMDCs across mLEC monolayers grown on the undersurface of Transwell filters, measured over a 5-h period by fluorescence plate reader. *$P < 0.05$, Mann–Whitney $U$-test. Data are the mean (center bar) ± s.e.m. (whiskers) (n = 4), one representative experiment of three. Source data are available for this figure.

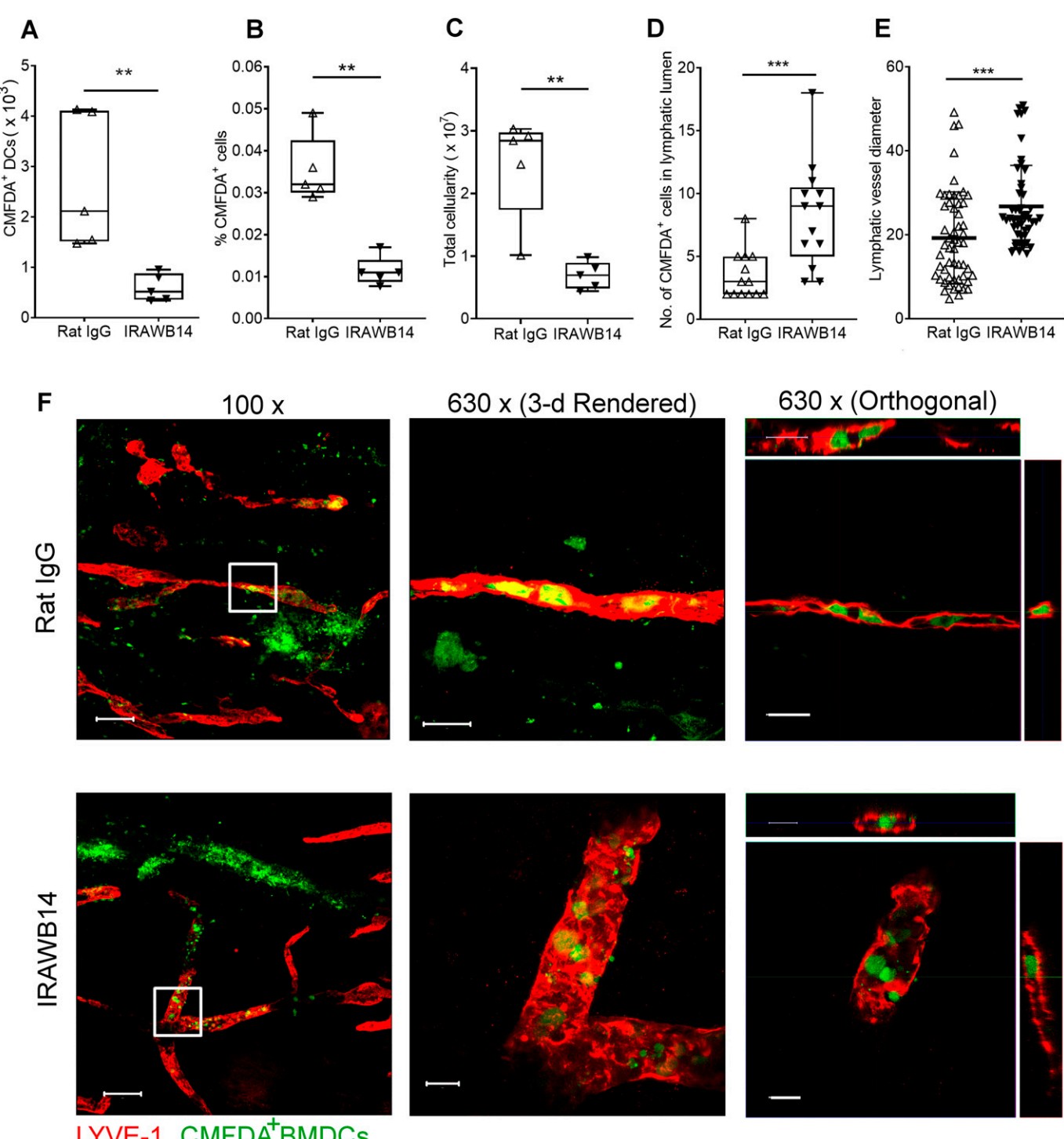

**Figure 10. Impaired lymphatic trafficking of DCs after mAb-induced enhancement of CD44–HA binding.**
**(A, B, C)** Effect of IRAWB14 or control rat IgG on recovery of LPS-matured CMFDA-labeled wild-type BMDCs from draining cervical LNs 24 h after intradermal injection into oxazolone-painted skin, as measured by flow cytometry and expressed as (A) numbers of CMFDA$^+$ BMDCs, (B) percentage of CMFDA+ cells and (C) total LN cellularity. **(D)** Numbers of BMDCs detected inside the lumen of dermal lymphatic vessels, expressed as a percentage of the total number of lymphatic vessel-associated BMDCs. **(E)** Lymphatic vessel diameter 24 h after adoptive transfer of IgG- and IRAWB14-treated BMDCs, five measurements/vessel, five vessels/mouse. **(F)** Entry of CMFDA-labeled IgG- and IRAWB14-treated wild-type BMDCs (green) into dermal afferent lymphatic capillaries immunostained with anti–LYVE-1 (red), following oxazolone skin painting. Panels show 3-D rendering of z-stacks at low magnification (left; 100×, scale bars = 100 μm), and higher magnification (center; 630×, scale bars = 20 μm) with orthogonal sections (right). **P < 0.01, ***P < 0.001, Mann–Whitney U-test. Data are the mean (center bar) ± s.e.m. (whiskers) (n = 5 mice), one representative experiment of three.

$CD44^{-/-}$ mice on C57BL/6 background were purchased from The Jackson Laboratory and maintained as a heterozygous colony. Experiments were carried out using $CD44^{-/-}$ and $CD44^{+/+}$ litter- and cage-mates aged 8–12 wk. C57BL/6 wild-type mice were purchased from Envigo RMS Inc.

### Genotyping

Ear notches from mice generated by $CD44^{-/+}$ × $CD44^{-/+}$ breeding were digested in 200 $\mu$l DirectPCR (Tail) lysis reagent (Viagen 102-T) supplemented with Proteinase K, 0.4 mg/ml (P2308; Sigma-Aldrich) for 16 h at 55°C, then heated to 85°C for 45 min before use in PCRs using a Bio-Rad Thermocycler T100. MyTaq Red Mix (Bioline) and primers (0.5 $\mu M$) were used for Touchdown PCR, denaturing at 94°C for 2 min initially, followed by 10 cycles of 94°C for 1 min, annealing at 65°C for 1 min (−0.5°C per cycle) and extending at 72°C for 1 min. Then, for a further 28 cycles, denaturing at 94°C for 1 min, annealing at 60°C for 1 min and extending at 72°C for 1 min, before a final 2 min incubation at 72°C. The following primers from The Jackson Laboratory Genotyping Protocol were used: oIMR1432 GGC GAC TAG ATC CCT CCG TT, oIMR1433 ACC CAG AGG CAT ACC AGC TG, oIMR6916 CTT GGG TGG AGA GGC TAT TC, and oIMR6917 AGG TGA GAT GAC AGG AGA TC, yielding PCR products of 280 bp for the null and 175 bp for the WT alleles, respectively. Products were electrophoresed on 1.5% agarose-Tris-Borate-EDTA gels.

### Antibodies

Rat anti-mouse LYVE-1 mAb C1/8 and mouse anti-human LYVE-1 mAb 6A were generated previously, using mouse LYVE-1 Fc and human LYVE-1 Fc, respectively, as immunogens, as were rabbit anti–LYVE-1 polyclonal antibodies (92, 93). Anti-CD44 mAb IRAWB14 hybridoma was kindly provided by Dr. Jayne Lesley and IM7 hybridoma was purchased from ATCC. Hybridomas were cultured in-house. Rat and mouse IgG fractions were purified from hybridoma supernatants using Protein G-Sepharose. Other antibodies were mouse anti-human CD44 mAb BRIC235 (International Blood Group Reference Library), mouse anti-human CD44 mAb F10.44.2-FITC (Abcam), anti-mouse CD11c-PE (12-0114-82 clone N418; eBioscience), anti-mouse MHCclII (I-A/I-E)-AF488 (Inc 5536323 clone 2G9; BioLegend), and anti-human ICAM3-APC (clone CBR-IC3/1; BioLegend Inc). All antibodies were used at 10 $\mu$g/ml for immunostaining and 50 $\mu$g/ml for function blocking. For flow cytometry, antibodies used were anti–CD11b-BUV395 (565976 clone M1/70; BD Biosciences), anti–MHCII (I-A/I-E)-eFluor 450 (48-5321-82 clone M5/114.15.2; eBioscience), anti–CD326 (EpCAM)-BV605-Brilliant Violet 605 (Inc 118227 clone G8.8; BioLegend), anti–CD45-Brilliant Violet 785 (103149 clone 30-F11BioLegend Inc.), anti–CD207 (Langerin)-PE (2-2075-82; eBioscience), anti–CD103-APC (17-1031-82 clone 2E7; Life Technologies Thermo Fisher Scientific), anti–F4/80-PE/Cyanine5 (123112 clone BM8; BioLegend Inc.), and anti–CD11c-PE/Cyanine7 (117318 clone N418; BioLegend Inc.), all used at 1 $\mu$g/ml. Biotinylated hyaluronan binding protein (bHABP), recombinant human versican G1 domain (bVG-1), from AMSBio (AMS.HKD-BC41), was used at 3 $\mu$g/ml, detecting with either streptavidin-conjugated Alexa Fluor 647 or streptavidin-conjugated Alexa Fluor Pacific Blue.

All secondary conjugates (Alexa Fluor 488, 546, 594, and 647) were purchased from Molecular Probes, Invitrogen. Irrelevant IgG isotype controls were purchased from R & D Systems.

Antibodies used for functional assays (both in vivo and in vitro) were tested for endotoxin contamination using a Pierce LAL Chromogenic Endotoxin Quantitation kit (88282; Thermo Fisher Scientific), according to the manufacturer's protocol, to ensure that endotoxin levels were less than 10 pg/ml.

### Cells

Primary mLECs and hLECs were prepared from freshly resected skin samples by immunoselection with LYVE-1 mAb and MACS beads (Miltenyi Biotec) and cultured on gelatin-coated tissue cultureware (0.1% in PBS) in EGM2MV media (Lonza), as described previously (26). Primary dermal fibroblasts were obtained by subculturing LYVE-1-negative cells for 12 d, to obtain rapidly growing cells that readily detach from substrate in the presence of EDTA (5 mM) in PBS.

BMDCs were extracted from tibia and fibula bones of euthanized mice, passed through a 70 $\mu$m cell strainer and cultured for 7 d in DC medium (RPMI 1640 with 10% FCS, kanamycin sulfate, MEM nonessential amino acids, sodium pyruvate, glutamine, and 2-mercaptoethanol (55 $\mu$M); all from Life Technologies), and supplemented with recombinant mouse GM-CSF and IL-4 (20 ng/ml, premium grade; Miltenyi Biotec). Monocytes were purified from PBMCs of healthy donors by positive immunoselection using anti-CD14–conjugated MACS beads (Miltenyi Biotec). MDDCs were generated by culturing monocytes for 5 d in DC medium, supplemented with recombinant human GM-CSF (50 ng/ml) and 10 ng/ml IL-4 (premium grade; Miltenyi Biotec). Non-adherent DCs were matured with 1 $\mu$g/ml LPS from *Salmonella abortus* (Sigma-Aldrich) where indicated.

### Measurement of in vivo DC trafficking

To study endogenous DC trafficking, 3% (wt/vol) oxazolone (4-ethoxymethylene-2 phenyl-2-oxazoline-5-one; E0753; Sigma-Aldrich) and 4 mg/ml FITC (Fluorescein isothiocyanate isomer 1, F7250; Sigma-Aldrich) in 95% aqueous ethanol were topically applied to the shaved abdomens (150 $\mu$l/mouse) of $CD44^{+/+}$ and $CD44^{-/-}$ mice aged 8 wk. Mice were euthanized after 24 h.

To measure trafficking of adoptively transferred BMDC, LPS-matured non-adherent BMDCs ($CD44^{-/-}$ or $CD44^{+/+}$ cells) were labeled with either Qtracker 655 (Q25021MP) or Qtracker 585 (Q25011MP) cell labeling kits, or CMFDA Cell Tracker Green (Invitrogen), according to the manufacturer's protocol. BMDCs were washed in PBS, mixed 1:1, then co-injected intradermally into the ears of wild-type mice, 2 × 10$^6$ cells total BMDCs per injection, at the same time as topical application of oxazolone, 3% (wt/vol) in 95% ethanol (vol/vol).

To assess trafficking of IRAWB14-treated cells, LPS-matured non-adherent wild-type BMDCs were labeled with CMFDA Cell Tracker Green (Invitrogen), following the manufacturer's protocol, then incubated with either rat IgG or IRAWB14 (100 $\mu$g/ml, in PBS) for 20 min on ice before injection into mouse ear dermis (2 × 10$^6$ cells in 50 $\mu$l per ear) at the same time as topical application of 3% oxazolone.

## Harvesting dermal DCs for cytospin

Ears were removed from $CD44^{-/-}$ and $CD44^{+/+}$ mice after euthanasia, then peeled into dorsal and ventral halves and cultured for 24 h (exposed dermis-side down) in 24-well dishes in RPMI 1640 supplemented with 10% FCS, penicillin–streptomycin and glutamine (Life Technologies), and mouse TNF$\alpha$ (50 ng/ml; R&D Systems). Cells which had crawled out from dermis were collected from the medium and applied to microscope slides by Cytospin (Shandon; Thermo Fisher Scientific).

## Flow cytometric analysis of LNs

LNs were cut into halves and digested at 37°C for 40 min in Collagenase D (11088882001; Roche), 1 mg/ml (wt/vol) in RPMI 1640, then mechanically disrupted through a 100 $\mu$m cell strainer. Cells were suspended in incubation buffer (PBS + 10% FCS, 0.01% azide), counted manually by hemocytometer and maintained on ice, and then incubated with Fixable Viability Dye eFluor780 (65-0865; eBioscience) for 15 min, after fixation and storage in 2% formaldehyde (vol/vol) at 4°C until analysis by flow cytometer (LSRII; BD Biosciences) and FlowJo software. Compensation was carried out using samples of single color-stained cells and fluorescence-minus-one controls.

## Flow cytometric analysis of cultured BMDCs

Non-adherent BMDCs were suspended in incubation buffer (PBS + 10% FCS, 0.01% wt/vol sodium azide) and maintained on ice for all incubation steps. Cells were first incubated with Fixable Viability Dye eFluor780 (65-0865; eBioscience) for 15 min, then with TruStain FcX Fc blocker (anti-mouse CD16/CD32 clone 93; BioLegend) for 15 min, before incubation with fluorescently conjugated antibodies for 20 min. Staining for HA was carried out after all other incubations, by fixing the cells in 2% formaldehyde (vol/vol) in PBS for 5 min, then incubating with recombinant biotin-labeled versican G1 domain (bVG1), 3 $\mu$g/ml for 40 min followed by streptavidin–Alexa Fluor 647 (S21374; Life Technologies) for 40 min. Cells were counted manually and analyzed using a flow cytometer (LSRII; BD Biosciences) and Flow-Jo software. Compensation was carried out using anti-rat/anti-hamster Ig CompBeads (BD 51-90-9000949) with negative control beads (BD 51-90-9001291), and fluorescence-minus-one controls. As a control for non-specific binding of bVG1, samples were treated with hyaluronidase (HAase, see below) before immunostaining.

## Immunofluorescence antibody staining of cells and tissues

Monolayers of mLECs or hLECs cultured in 8-chamber slides (BD Falcon) were fixed in paraformaldehyde (1% wt/vol in PBS, pH 7.4) for 5 min, and then washed in PBS before applying primary antibodies in blocking buffer (PBS + 1% BSA + 10% FCS). Cells were incubated at room temperature for 45 min, followed by washing and further incubation for 30 min with Alexa Fluor secondary antibodies, before mounting in Vectashield+DAPI (H-1200; Vector Laboratories) and viewing on a Zeiss LSM 780 confocal microscope. Images were captured by sequential scanning, with no overlap in detection of emissions from each fluorophore, using either a 10X/ 0.3 DIC M27 Plan-Apochromat, or 40X/1.1 W Korr UV-Vis-IR LDC-Apochromat, or 63X/1.4 oil Plan-Apochromat objectives.

To visualize the HA glycocalyx of $CD44^{+/+}$ and $CD44^{-/-}$ BMDCs, non-adherent LPS-matured cells in suspension were labeled with CMFDA green tracker dye where indicated, fixed in 2% PFA (vol/vol in PBS), 10 min room temperature, then washed in PBS and incubated in blocking buffer (5% FCS and 1% BSA in PBS) for 20 min. Immunostaining was carried out in blocking buffer, with cells in suspension. For counterstaining with phalloidin, cells were permeabilized with Triton X-100 (0.1% in PBS), 10 min, before incubation with Rhodamine–Phalloidin, 1:200 (R415; Invitrogen Thermo Fisher Scientific) 30 min room temperature. Imaging was carried out using a Zeiss LSM 880 inverted microscope with Airyscan detector and 63X/1.4 oil Plan-Apochromat objective, using default image reconstruction parameters in Zeiss Zen software. The thickness of HA pericellular coats of $CD44^{+/+}$ and $CD44^{-/-}$ BMDCs was measured using Image J, at 30 positions around individual cells.

For whole-mount staining, mouse dermis was fixed in 1% paraformaldehyde for 1 h, washed in PBS-Triton X-100 (0.3% vol/vol), blocked with BSA (1% wt/vol) and FCS (10% vol/vol), and incubated with primary antibodies at 4°C overnight and fluorescently conjugated secondary antibodies for 2 h at room temperature. Tissue samples were then mounted in Vectashield (H-1000; Vector Laboratories) and viewed using a Zeiss LSM 780 upright confocal microscope, using sequential scanning and either a 10X/0.3 DIC M27 Plan-Apochromat, 40X/1.1 W Korr UV-Vis-IR LDC-Apochromat, or 63X/1.4 oil Plan-Apochromat objective.

## Removal of surface HA from BMDCs and tissue

BMDCs or tissue were incubated in either PBS alone or with *Streptomyces hyalurolyticus* HAase (15 U/ml; Sigma-Aldrich), for 2 h at 37°C in PBS, then washed in PBS.

## Quantitation of LYVE-1⁺ endothelial cup formation

Primary mLECs or hLECs were seeded in eight-chamber slides and cultured until confluent. CMFDA fluorescently labeled LPS-matured BMDCs or MDDCs were preincubated with either anti-CD44 mAbs or control IgG 30 min as appropriate, before addition of DCs (0.1 × 10⁶ per chamber). Cells were incubated at 37°C for 3 h then non-adherent DCs were removed by gentle washing with PBS, before immunostaining as detailed above. Adherent DCs from 10 fields of view per chamber were counted and scored as either associated with or independent of LYVE-1⁺ transmigratory cups.

## Adhesion and transmigration assays

To quantify adhesion, confluent monolayers of primary mLECs in gelatin-coated 24-well dishes were layered with 5 × 10⁵ fluorescently labeled LPS-matured BMDCs per well and incubated at 37°C for 3 h. The total numbers of BMDCs present were then measured in a fluorescence plate reader (Synergy HT; Bio-Tek), followed by gentle rinsing (three times with PBS) to remove non-adherent BMDCs, before re-measuring fluorescence and calculating the percentage of adherent cells. To assess the effect of CD44 clustering/activation, CMFDA-labeled BMDCs were preincubated

(30 min, 37°C) with CD44 mAb IRAWB14 or control Ig before applying to mLEC monolayers, maintaining the presence of Ab throughout.

For measurement of transmigration, primary mLECs were seeded onto the underside of gelatin-coated Fluoroblok cell culture inserts (3 $\mu$m pore size; BD Falcon), or primary fibroblasts were seeded on the upper surface of such inserts, culturing in companion plates until fully confluent. Wild-type LPS-matured BMDCs were then fluorescently labeled with CMFDA green tracker dye and pre-incubated (37°C, 30 min) with IRAWB14 or control Ig before application to the upper chambers of inserts ($5 \times 10^5$ BMDCs per well). LPS-matured BMDCs prepared from $CD44^{+/+}$ and $CD44^{-/-}$ littermates were also fluorescently labeled but applied directly to the upper chambers of inserts ($5 \times 10^5$ BMDCs per well). To assess the contribution of LYVE-1 to diapedesis of $CD44^{+/+}$ and $CD44^{-/-}$ BMDCs, the function blocking anti–LYVE-1 antibody mAb 2125 or control rat IgG was applied to mLEC 30 min before addition of BMDCs, and maintained throughout the course of the experiment. In each case, BMDCs transiting to the lower chambers were quantified over a 5-h period at 37°C using a fluorescence plate reader (Synergy HT; Bio-Tek) as described previously ([26]).

### Imaging of DC motility on LEC monolayers

mLECs or hLECs were plated on 0.1% gelatin-coated glass-bottomed wells (Ibidi) and cultured until confluent. Where indicated, hLECs were incubated for 30 min at 37°C with mouse anti–LYVE-1 mAb (clone 6A, 10 $\mu$g/ml), conjugated to DyLight 650 according to the manufacturer's instructions (Thermo Fisher Scientific), 30 min, then washed in EGM2MV medium. LPS-matured MDDCs were either pre-labeled with CMFDA Cell Tracker Green (Invitrogen), following the manufacturer's protocol, or with fluorescently conjugated anti-CD44 mAb clone F10.44.2 and anti-ICAM3, washing in EGM2MV medium before application to hLEC monolayers. Where indicated, CMFDA-labeled MDDCs were preincubated with either control IgG or anti-CD44 mAbs IM7, BRIC235, or F10.44.2, 10 $\mu$g/ml, 30 min before applying directly to hLEC monolayers. $CD44^{+/+}$ and $CD44^{-/-}$ BMDCs were pre-labeled with either CMFDA Cell Tracker Green or CMTPX Cell Tracker Red, washed in EGM2MV medium and then applied to mLEC monolayers.

Time-lapse images were acquired using a Zeiss Cell Observer inverted spinning disc confocal microscope, equipped with a Hamamatsu Flash v2 CMOS camera and a 20× 0.8 NA Plan-Apochromat objective, at 37°C in the presence of 5% $CO_2$. Cells were allowed to equilibrate at the microscope for a minimum of 15 min before imaging. Spinning disc images were deconvolved using Huygens Professional (version 19.10, Scientific Volume Imaging) using the Classical Maximum Likelihood method and default parameters as determined from the image meta data.

### Quantitation of DC hyaluronan levels

A hyaluronan competitive ELISA kit (Cat. no. K-1200-1) was purchased from Eschelon Biosciences and the manufacturer's protocol was followed. BMDCs were prepared from $CD44^{+/+}$ and $CD44^{-/-}$ mice as described above, culturing for 24 h either alone or with LPS (1 $\mu$g/ml). Cells were counted, lysed in lysis buffer (50 mM Tris, pH 7.5, 100 mM NaCl, 1% NP-40, and 1 mM EDTA) and debris removed by

centrifugation (11,000$g$, 5 min) before application to the assay. Supernatants were applied directly to the assay without further treatment. HA concentration was expressed as pg/cell.

### HA synthase quantitation by Western blotting

Immature and LPS-matured BMDCs were lysed (98°C, 5 min) in 1× NuPAGE LDS sample buffer (Thermo Fisher Scientific) containing 2-mercaptoethanol (5%, vol/vol) for reducing SDS–PAGE. To obtain a positive control for immunoblotting, skin was excised from the back of euthanized C57BL/6 mice, after treatment with hair removal cream (Veet), then minced in PBS containing Complete EDTA-free protease inhibitor (Merck) on ice, before low-speed centrifugation. Supernatant was discarded, and sedimented tissue homogenized in 1× NuPAGE LDS sample buffer, lysed (98°C, 5 min) then centrifuged (11,000$g$, 5 min), from which supernatant was retained. All samples were subjected to sonication (Bioruptor pico, Diagenode) to shear genomic DNA, then electrophoresed on Novex 4–12% Tris-Glycine Mini Protein gels followed by transfer to Immobilon-Fl PVDF membranes (Millipore). Blots were blocked by LI-COR blocking buffer at room temperature, then probed with mouse anti-GAPDH (clone 6C5; Life Technologies Ltd) and either rabbit anti-HAS1 (Cat. no. A10453, supplied by 2B Scientific; ABclonal), anti-HAS2 (Cat. no. A9897; Abclonal), or rabbit anti-HAS3 (Cat. no. PA5-100552; Thermo Fisher Scientific) in blocking buffer (4°C, overnight). After washing with PBS 0.1% (vol/vol) Tween 20 for 4 × 5 min, the membrane was incubated with the appropriate IRdye 800 and IRdye 700 conjugates for quantitative imaging using a LI-COR Odyssey scanner and Image Studio software.

### Data and statistical analyses

Data were analyzed using Excel (Microsoft) and the Mann–Whitney U test was used to compare data sets throughout this study, unless otherwise stated, using Graph Pad Prism. $P < 0.05$ was considered significant. Microscopy images were quantitated using Image J and Imaris (Bitplane).

### Online supplemental material

Fig S1 demonstrates that CD44 deficiency does not affect expression of DC activation markers MHC class II, CD80, or CD86. Fig S2 shows the specificity of the bVG1 probe for HA in BMDCs. Fig S3 reveals reduced levels of HA in the dermis of $CD44^{-/-}$ mice. Fig S4 demonstrates that genetic deletion of CD44 does not affect expression of HAS enzymes in BMDCs. Fig S5 illustrates that disrupting CD44-HA interactions using CD44 function-blocking mAbs prevents formation of transmigratory cups in lymphatic endothelium. Fig S6 shows impaired trafficking of endogenous dermal DC subsets to draining LNs in CD44-deficient mice. Fig S7 demonstrates reduced lymphatic trafficking of adoptively transferred $CD44^{-/-}$ BMDCs, as a percentage of total number of cells in draining LNs. Fig S8 confirms no IRAWB14-induced loss of BMDC viability and shows that mAb-induced potentiation of CD44:HA interactions impairs lymphatic trafficking.

All Videos are spinning disc confocal microscopy time-lapse movies. Videos 1–Videos 6 show the increase in MDDC motility on monolayers of hLECs in the presence of the function-blocking

anti-CD44 mAbs indicated. Video 7 shows an individual CD44$^{+/+}$ BMDC adhering to a monolayer of mLECs, whereas an adjacent CD44$^{-/-}$ BMDC is only loosely attached and continues to crawl.

# Supplementary Information

# Acknowledgements

The authors thank P Sopp, C Waugh, K Clark, and S-A Clark at the Weatherall Institute of Molecular Medicine Flow Cytometry Facility for their excellent technical assistance and advice. This work was supported by the UK Medical Research Council (MRC) through MRC Human Immunology Unit core funding (DG Jackson) and the Wolfson Imaging Centre (BC Lagerholm), MRC Weatherall Institute of Molecular Medicine.

## Author Contributions

LA Johnson: conceptualization, resources, data curation, formal analysis, investigation, visualization, methodology, project administration, and writing—original draft, review, and editing.
S Banerji: investigation, methodology, and writing—review and editing.
BC Lagerholm: resources, investigation, and methodology.
DG Jackson: conceptualization, funding acquisition, resources, project management, and writing—review and editing.

## Conflict of Interest Statement

The authors declare that they have no conflict of interest.

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
