## [Reviewer comments · Life Science Alliance]

Life Science Alliance

Dendritic cell entry to lymphatic capillaries is orchestrated by CD44 and the hyaluronan glycoalyx

Louise Johnson, Suneale Banerji, Christoffer Lagerholm, and David Jackson

DOI: <https://doi.org/10.26508/lsa.20200908>

Corresponding author(s): Louise Johnson, University of Oxford Weatherall Institute of Molecular Medicine, University of Oxford and David Jackson, University of Oxford Weatherall Institute of Molecular Medicine, University of Oxford

Review Timeline:

Submission Date:	2020-09-16
Editorial Decision:	2020-10-10
Revision Received:	2021-01-29
Editorial Decision:	2021-02-18
Revision Received:	2021-02-22
Accepted:	2021-02-22

Scientific Editor: Shachi Bhatt

Transaction Report:

October 10, 2020

Re: Life Science Alliance manuscript #LSA-2020-00908-T

Mrs. Louise Anne Johnson
University of Oxford Weatherall Institute of Molecular Medicine, University of Oxford
MRC Human Immunology Unit
John Radcliffe Hospital
Headington
Oxford OX3 9DS
United Kingdom

Dear Dr. Johnson,

Thank you for submitting your manuscript entitled "Dendritic cell entry to lymphatic capillaries is regulated by CD44 and the hyaluronan glycoalyx" to Life Science Alliance. The manuscript was assessed by expert reviewers, whose comments are appended to this letter.

As you can see from the reviewer reports below, the reviewers are quite enthusiastic about the study, but have pointed out some concerns that should be addressed prior to further consideration of this manuscript at Life Science Alliance. Thus, we encourage you to submit a revised manuscript addressing all of the reviewers' comments.

Thank you for this interesting contribution to Life Science Alliance. We are looking forward to receiving your revised manuscript.

Sincerely,

Shachi Bhatt, Ph.D.
Executive Editor
Life Science Alliance
<https://www.life-science-alliance.org/>
Tweet @SciBhatt @LSAJournal

- A letter addressing the reviewers' comments point by point.
- An editable version of the final text (.DOC or .DOCX) is needed for copyediting (no PDFs).
- High-resolution figure, supplementary figure and video files uploaded as individual files: See our detailed guidelines for preparing your production-ready images, <https://www.life-science-alliance.org/authors>
- Summary blurb (enter in submission system): A short text summarizing in a single sentence the study (max. 200 characters including spaces). This text is used in conjunction with the titles of papers, hence should be informative and complementary to the title and running title. It should describe the context and significance of the findings for a general readership; it should be written in the present tense and refer to the work in the third person. Author names should not be mentioned.

B. MANUSCRIPT ORGANIZATION AND FORMATTING:

Reviewer #1 (Comments to the Authors (Required)):

The paper reports in a very detailed manner how dendritic cell CD44 interacts with LYVE-1 via hyaluronan link during their interaction with afferent lymphatic endothelial cells. The hyaluronan link has been suspected in this interaction but has not been demonstrated earlier. In this aspect, the paper brings new knowledge to the field, especially to those interested in leukocyte trafficking. The paper is a carefully constructed thorough investigation and the data support the conclusions.

Specific comment:

Combined data of all experiments should be shown in Figures. Representatives are fine for images and gels, but the bars should represent combined results.

Minor comment: As it is known that also lymphocytes use CD44 to interact with lymphatic endothelium. The authors could speculate on the basis of the known knowledge, whether the DC type of interaction would also be possible for lymphocytes or could it be that macrophage mannose receptor is a more important interaction partner for lymphocyte CD44.

Reviewer #2 (Comments to the Authors (Required)):

These authors have examined the role of CD44 in the structure of the hyaluronan (HA) glycolyx surrounding dendritic cells and entry of these cells into lymphatic capillaries. They report that a) CD44 dynamically controls the density of the hyaluronan glycolyx, b) CD44 is required for DC adhesion and transmigration, and c) that these pericellular CD44/HA interactions regulate the efficiency of DC trafficking to lymph nodes.

Points a) and b) above are well established in a series of carefully controlled and beautifully presented experiments. However, point c) - that CD44-dependent effects on pericellular HA are responsible for DC trafficking in vivo - is less well supported. There are also technical limitations in the approach, including a reliance on differential effects of anti-CD44 antibody clones, that make aspects of the work difficult to interpret.

Otherwise this is a beautiful, well-written and thoughtful work that uses cutting edge methods to put a fresh perspective on CD44, pericellular HA, and DC migration.

Major concerns:

1. There is a large literature on CD44- effects on DC migration that should be cited and discussed including: PMID: 23603511, PMID: 10583604, and PMID: 9166413, PMID: 20574435). The statement in the discussion that "we have provided the first evidence that CD44 plays an active role in DC trafficking"" should be struck in light of PMID: 9166413.

2. A role for pericellular HA in lymph node homing of DCs is not interrogated directly. Most of the experiments in question use either CD44-/- deficient cells or anti-CD44 antibodies. However, CD44 has numerous other ligands that impact DC migration (PMID: 23603511; PMID: 11696588) and the loss of CD44 affects many aspects of cell behavior. The use of hyaluronidase (Figure 7) in vitro lends some support to this model but this treatment would affect all of the cells in culture and therefore is not specific to DC and it is not used in vivo. Given these limitations, a role for pericellular CD44/HA interactions in DC trafficking to lymph nodes unfortunately can not be said to be proven by these data.

To demonstrate this, it would be necessary to genetically over-express or otherwise synthetically engineer pericellular HA on DC in the absence or presence of CD44.

3. The finding that DC do not express HAS3 contrasts sharply with previous reports that HAS3 is the primary HA synthase expressed by immune cells (PMID: 28987865) including dendritic cells (PMID: 20228832, PMID: 12370364) Moreover, it conflicts with reports that HAS3 is primary cell surface HA synthase (PMID: 24057227). Given this discrepancy it would be important to repeat this

part of the experiment in Figure 2 with a control known to express HAS3.

4. The authors rely heavily on the differential effects of various CD44 antibodies to test various aspects of their model. However, the effects of these antibodies are complex and difficult to control for. For example clone IRAWB14 increases HA binding (its use here) but it also enhances CD44 shedding (PMID: 10211879) and may promote proliferation and survival (PMID: 20079666). In figure 8 it would be important to control for these effects by assessing viability and proliferation over the time frame of the experiment.

Reviewer #3 (Comments to the Authors (Required)):

In this study, Johnson et al. report on a novel role for dendritic cell (DC)-expressed CD44 in regulating DC adhesion and transmigration into afferent lymphatics. Performing experiments with CD44-deficient bone marrow-derived (BM)-DCs as well as with different CD44-directed antibodies the authors show that by binding hyaluronan, CD44 regulates the density of the DC's hyaluronan glycoocalyx and CD44/hyaluronan-mediated interactions with LYVE-1 expressing lymphatic endothelial cells (LECs). In absence of CD44 adoptively transferred BM-DCs displayed reduced migration to draining lymph nodes (dLNs). Surprisingly enhancement of the hyaluronan glycoocalyx density (mediated by a CD44-clustering antibody) did not enhance but rather reduce DC migration, presumably by causing enhanced adhesion and slow-down of DC migrating in lymphatic capillaries.

Major points:

1. In their adoptive transfer studies the author recovered less CD44^{-/-} DCs from draining LNs but significantly more from the ear skin. It is not clear to me what the quantification in Fig. 6B shows (the y-axis refers to an MFI.). In case the authors indeed quantified tissue whole mounts, as indicated in the Figure Legends, it would be nice if representative sections were shown. Considering that in adoptive transfers typically only a small fraction (less than 1%) of DCs actually arrive and are quantified by FACS in the draining LNs, it is somewhat surprising that numbers of CD44^{-/-} DCs were almost doubled in the quantified dermis. Perhaps a better way of quantifying cells remaining in the skin would be flow cytometry of enzymatically digested skin (in analogy to the LN analysis).

2. In the DC transmigration assays shown in Figure 7: it would be nice if these findings (involvement of CD44:HA-LYVE-1 axis) could be confirmed by performing transmigration assays with LYVE-1-deficient LECs. According to the hypothesis / findings presented in this study, one would expect that CD44^{-/-} and WT DCs would migrate equally well across LYVE-1-deficient LEC monolayers.

3. The authors mention former studies analysing Langerhans cell migration in CD44^{-/-} mice, which apparently yielded conflicting results "due to the abundant expression of CD44 in the tissue". Have the authors performed any FITC painting experiments themselves to address the role of CD44 in the migration of endogenous DCs in this setup (either in complete KOs or in BM-chimeras)? A certain limitation of their study obviously is that it only migration data from adoptively transferred BM-DCs are provided.

4. Figure 8/9: While I can understand how anti-CD44 mAB IRAWB14 treatment may enhance DC adhesion, I do not entirely follow why this treatment would lead to reduced DC transmigration and

overall migration to dLNs (rather than enhancing this process). I noticed that the control antibody chosen does not bind DCs - a better control in the assay likely would be an antibody that also attaches to the cells (e.g. to CD44 or any other molecule, but without functional consequences).

Minor points:

1. Suppl. 2: please provide FACS quantifications, in addition to the images
2. A general remark: Please provide representative FACS plots for all quantifications shown (e.g. Fig. 8A,B, Suppl. Figure 1)
3. Fig. 6, adoptive transfer experiments: please also show LN cellularity and percentage % data of the DC migration defect
4. Figure 9C: The fact that the lymphatic vessel diameter expands upon injection of DCs is surprising and could also be explained by other mechanism than "occlusion of lymph flow due to trapped BM-DCs". E.g. it could be that this antibody induces inflammatory signals in the ear skin (I assume the authors cannot be certain that it remains on the DCs and doesn't bind to other CD44-expressing cells). Please include further possible explanations or else provide more evidence suggesting that lymph flow is indeed reduced.

Shachi Bhatt, Ph.D
Executive Editor
Life Science Alliance
950 3rd Avenue, Floor 2
New York, NY, 10022, USA

Manuscript #LSA-2020-00908-T

29th January, 2021

“Dendritic cell entry to lymphatic capillaries is orchestrated by CD44 and the hyaluronan glycocalyx.”

Dear Dr. Bhatt,

Thank you very much for providing us with the opportunity to resubmit our manuscript, which we have revised with further experiments and re-writing, in response to the reviewers' critiques. We would like to thank the reviewers for their thorough evaluation of our manuscript and their helpful comments and criticisms. We have now addressed all of their concerns and believe that it has greatly strengthened our manuscript. Below are our point-by-point responses (in black) to their comments (in blue). Revisions to the text in the manuscript are shown in red.

Reviewer #1 (Comments to the Authors (Required)):

The paper reports in a very detailed manner how dendritic cell CD44 interacts with LYVE-1 via hyaluronan link during their interaction with afferent lymphatic endothelial cells. The hyaluronan link has been suspected in this interaction but has not been demonstrated earlier. In this aspect, the paper brings new knowledge to the field, especially to those interested in leukocyte trafficking. The paper is a carefully constructed thorough investigation and the data support the conclusions.

Specific comment:

Combined data of all experiments should be shown in Figures. Representatives are fine for images and gels, but the bars should represent combined results.

We greatly appreciate the fact that this reviewer recognizes the important contribution that our study makes to the field of leukocyte trafficking.

We have combined data from replicate experiments where possible and show these in our revised manuscript, specifically in figures 1E, 1G, 3D, 3E, 4, 5C, 7D, 10D and 10E, with the figure legends revised accordingly. However, all of the *in vivo* experiments detailed in the manuscript use litter- and cage-mates to allow clearer comparisons and combat natural variations. Also, flow cytometer settings were different between each FACS experiment as they

were set according to the fluorescence-minus-one and compensation bead controls that were immunostained in parallel with the samples. Similarly, the *in vitro* experiments we describe used primary cells, which vary between human donors and mice. Thus, data from certain replicate experiments cannot be combined, despite the trends and the statistical significance across all replicates of experiments. We include at least one replicate of such experiments in an attached figure to this letter.

Minor comment: As it is known that also lymphocytes use CD44 to interact with lymphatic endothelium. The authors could speculate on the basis of the known knowledge, whether the DC type of interaction would also be possible for lymphocytes or could it be that macrophage mannose receptor is a more important interaction partner for lymphocyte CD44.

The reviewer raises a valid point and we include a new paragraph to discuss this on page 17 of our revised manuscript.

Reviewer #2 (Comments to the Authors (Required)):

These authors have examined the role of CD44 in the structure of the hyaluronan (HA) glyocalyx surrounding dendritic cells and entry of these cells into lymphatic capillaries. They report that a) CD44 dynamically controls the density of the hyaluronan glyocalyx, b) CD44 is required for DC adhesion and transmigration, and c) that these pericellular CD44/HA interactions regulate the efficiency of DC trafficking to lymph nodes.

Points a) and b) above are well established in a series of carefully controlled and beautifully presented experiments. However, point c) - that CD44-dependent effects on pericellular HA are responsible for DC trafficking *in vivo* - is less well supported. There are also technical limitations in the approach, including a reliance on differential effects of anti-CD44 antibody clones, that make aspects of the work difficult to interpret.

We would like to thank the reviewer for their compliments on our experiments to support points a) and b). However, with respect, we consider that our data do provide substantial support for the conclusion that CD44, through its retention of the HA glyocalyx, indeed plays an integral role in DC trafficking *in vivo*. It should be noted that we do not claim such CD44-dependent effects are responsible for DC trafficking as described by the reviewer, but rather they are important mediators of the process. Our conclusions are based not only on studies with CD44 antibodies but also on studies using DCs isolated from CD44^{-/-} mice and their wild-type litter-mate controls. Furthermore, the CD44 monoclonal antibodies we used (IRAWB14, IM7, BRIC235) are well-characterized reagents whose functional effects on HA binding have been extensively documented by their originators and others in numerous different contexts, both *in vitro* and *in vivo*. For examples, please see PMID: 7688309 (Lesley et al 1993); PMID: 1730918 (Lesley et al 1992); PMID: 16497973 (Bonder et al 2006); PMID:10211879 (Mikecz et al 1999); PMID:12750406 (Kato et al 2003); PMID: 10583604 (Brennan et al 1999; PMID: 7535820 (Hamann et al 1995); PMID: 7561101 (Liao et al 1995); PMID: 23384599 (Teh et al 2013); PMID: 27679982 (Bano et al 2016); and PMID: 12801931 (Sugahara et al 2003).

Nevertheless, we agree that it would be more accurate to state that our studies with the IRAWB14 mAb indicate it is feasible that CD44 could regulate DC trafficking *in vivo*, for example through the known modulation of its binding affinity during inflammation and its ability to cluster the HA glyocalyx. In line with the reviewer's concerns, we have softened the title of our manuscript to "Dendritic cell entry to lymphatic capillaries is orchestrated by CD44 and the hyaluronan glyocalyx" and have also amended a sentence in the final paragraph of the discussion to, "we have demonstrated that modulation of HA glyocalyx density through CD44 clustering and activation can regulate the efficiency of DC docking and entry to lymphatics, and may possibly influence the rate of DC trafficking to draining LNs *in vivo*."

Otherwise this is a beautiful, well-written and thoughtful work that uses cutting edge methods to put a fresh perspective on CD44, pericellular HA, and DC migration.

Thank you very much!

Major concerns:

1. There is a large literature on CD44- effects on DC migration that should be cited and discussed including: PMID: 23603511, PMID: 10583604, and PMID: 9166413, PMID: 20574435). The statement in the discussion that "we have provided the first evidence that CD44 plays an active role in DC trafficking" should be struck in light of PMID: 9166413.

PMID: 23603511 (Salmi et al, 2013) and PMID: 10583604 (Brennan et al 1999) demonstrate roles of CD44 in T-cell homing to draining LNs, while PMID: 9166413 (Weiss et al 1997) and PMID: 20574435 (Miaw et al 2010) focus on Langerhans cell migration, with Weiss et al primarily focusing on the expression of CD44 splice variants. We cite Miaw et al and Mummert et al (2004) on page 14 of the manuscript, as these references were the most relevant to DC trafficking, and we now also cite Salmi et al (PMID: 23603511) in our revised manuscript (page 17) as requested. In addition, we have amended the statement at issue in the discussion to "we have provided new evidence that CD44 plays an active role in DC trafficking" as we believe that our study provides direct and detailed evidence that CD44 plays a role in initial adherence of DCs to lymphatic endothelium by presenting an HA surface glycocalyx.

2. A role for pericellular HA in lymph node homing of DCs is not interrogated directly. Most of the experiments in question use either CD44^{-/-} deficient cells or anti-CD44 antibodies. However, CD44 has numerous other ligands that impact DC migration (PMID: 23603511; PMID: 11696588) and the loss of CD44 affects many aspects of cell behavior. The use of hyaluronidase (Figure 7) *in vitro* lends some support to this model but this treatment would affect all of the cells in culture and therefore is not specific to DC and it is not used *in vivo*. Given these limitations, a role for pericellular CD44/HA interactions in DC trafficking to lymph nodes unfortunately can not be said to be proven by these data.

Respectfully, we disagree with the reviewer on this point. In a previous publication (Johnson et al 2017) which we cite in our current manuscript, we treated BMDCs with hyaluronidase and the pharmacological hyaluronan synthase inhibitor 4-methylumbelliferone prior to adoptive transfer into the dermis of recipient (untreated) mice and demonstrated impaired lymphatic trafficking of these cells to draining LNs. In the same publication, we showed that treatment of DCs with hyaluronidase disrupted adhesion to lymphatic endothelium *in vitro*, whereas similar treatment of lymphatic endothelial cells had no such effect. Furthermore, in this present manuscript, only BMDCs alone (and not lymphatic endothelial cells) were treated with hyaluronidase and thus not all of the cells in culture were affected – as asserted by the reviewer. Specifically, as shown in Fig. 8B, we found that "hyaluronidase reduced the adhesion of CD44^{+/-} BMDCs to the same level as that of CD44^{-/-} BMDCs, confirming that CD44:HA interactions are responsible for supporting such DC-LEC adherence, rather than other ligands of CD44". For balance, in our revised manuscript, we also include references to macrophage mannose receptor (PMID: 23603511) and osteopontin (PMID: 11696588), page 10 as other potential CD44 ligands. In addition, we include new data within a further panel (Fig. 8C) to show that free high molecular weight HA reduces adhesion of BMDCs to mLECs. Again, this supports our hypothesis that engagement of the HA glycocalyx is key to DC endothelial adhesion and that the DC HA glycocalyx is stabilized almost exclusively by CD44. Indeed, the single most dramatic effect of CD44 deletion in DCs is almost total loss of the HA glycocalyx.

To demonstrate this, it would be necessary to genetically over-express or otherwise synthetically engineer pericellular HA on DC in the absence or presence of CD44.

We feel that the arguments presented above and the inclusion of the new data from the HA competition studies (Fig 8C) make such an elaborate experiment unnecessary. Moreover, the genetic over-expression of HAS as suggested by the reviewer would likely generate DCs with a hyper-dense glycocalyx and grossly altered LYVE-1 binding capacity. In addition, the increase in free HA secretion that would likely manifest in such DCs would almost certainly interfere with LYVE-1 binding and potentially complicate interpretation of the results in the proposed studies.

3. The finding that DC do not express HAS3 contrasts sharply with previous reports that HAS3 is the primary HA synthase expressed by immune cells (PMID: 28987865) including dendritic cells (PMID: 20228832, PMID: 12370364) Moreover, it conflicts with reports that HAS3 is primary cell surface HA synthase (PMID: 24057227). Given this discrepancy it would be important to repeat this part of the experiment in Figure 2 with a control known to express HAS3.

We agree that it is important to repeat this experiment with other antibodies and positive controls to confirm this finding and apologize for the oversight in our previous submission.

We now include new data in a revised Fig. 2 and Supplementary Fig. S4, using a second antibody against HAS3 and a positive control of lysate prepared from mouse skin, and we thank the reviewer for alerting us to this. As before, we were unable to detect HAS3 in DC lysates and measured no difference in levels between immature and mature BMDCs from either CD44^{+/+} or CD44^{-/-} mice. It is interesting to note that expression of HAS at the protein level in DCs has not been examined in the previous studies mentioned by the reviewer. In particular, Homann et al, 2018 (PMID: 28987865) reported HAS3 expression in vascular smooth muscle cells rather than immune cells. Furthermore, studies by Bollyky et al, 2010 (PMID: 20228832) and Mummert et al 2002 (PMID: 12370364) both evaluated message levels of HAS rather than expression at the protein level, and Törrönen et al, 2013 (PMID: 24057227) examined HAS protein on fibroblasts and keratinocytes. Hence, with respect, surprisingly little is known about HAS3 expression in mouse DCs at the protein level.

4. The authors rely heavily on the differential effects of various CD44 antibodies to test various aspects of their model. However, the effects of these antibodies are complex and difficult to control for. For example clone IRAWB14 increases HA binding (its use here) but it also enhances CD44 shedding (PMID: 10211879) and may promote proliferation and survival (PMID: 20079666). In figure 8 it would be important to control for these effects by assessing viability and proliferation over the time frame of the experiment.

To complement our experiments using CD44-deficient mice, we have indeed used various anti-CD44 mAbs, all of which have been used and characterized extensively in respectable scientific studies in the past (including, but by no means limited to, those by Jayne Lesley, Paul Kincade, Ellen Pure and Timothy Springer). Our *in vitro* experiments involved incubating DCs with IRAWB14 for 3 h, whereas Mikecz et al (1999) incubated synovial and bone marrow stromal cells with immobilized IRAWB14 for 16 h. Importantly, these authors also state, "Although treatment with IRAWB14 also causes significant loss of CD44 molecules (ref. 33 and present report), due to enhanced HA-binding activity of receptors that remain unaffected by shedding, the ligand-binding capacity of CD44 can be restored on the cell surface and eventually increased on a per-cell basis. Thus, upon repeated *in vivo* administration of IRAWB14, its beneficial effect (i.e., induction of CD44 shedding) is defeated by the strong stimulatory action of this mAb on CD44-HA binding." The proliferation mentioned in Baaten et al, 2010 (PMID 20079666) was in OT-II cells responding to antigen, rather than bone marrow-derived DCs, which do not proliferate, and the authors state, "However, IRAWB 14 did not promote proliferation of *in vitro* activated WT OT-II cells (data not shown), and other studies

support the concept that ligation of CD44 without TCR signaling does not promote division of T cells (Marhaba *et al.*, 2006).” In our *in vivo* trafficking experiments, BMDCs were exposed to IRAWB14 for 24 h, following which we confirmed that there is no loss of viability and also that IRAWB14 is still bound to the BMDC surface after 24 h incubation *in vitro*. We include these new data in Supplementary Fig. S8 in the revised manuscript.

Reviewer #3 (Comments to the Authors (Required)):

In this study, Johnson *et al.* report on a novel role for dendritic cell (DC)-expressed CD44 in regulating DC adhesion and transmigration into afferent lymphatics. Performing experiments with CD44-deficient bone marrow-derived (BM)-DCs as well as with different CD44-directed antibodies the authors show that by binding hyaluronan, CD44 regulates the density of the DC's hyaluronan glycocalyx and CD44/hyaluronan-mediated interactions with LYVE-1 expressing lymphatic endothelial cells (LECs). In absence of CD44 adoptively transferred BM-DCs displayed reduced migration to draining lymph nodes (dLNs). Surprisingly enhancement of the hyaluronan glycocalyx density (mediated by a CD44-clustering antibody) did not enhance but rather reduce DC migration, presumably by causing enhanced adhesion and slow-down of DC migrating in lymphatic capillaries.

Major points:

1. In their adoptive transfer studies the author recovered less CD44^{-/-} DCs from draining LNs but significantly more from the ear skin. It is not clear to me what the quantification in Fig. 6B shows (the y-axis refers to an MFI..). In case the authors indeed quantified tissue whole mounts, as indicated in the Figure Legends, it would be nice if representative sections were shown. Considering that in adoptive transfers typically only a small fraction (less than 1%) of DCs actually arrive and are quantified by FACS in the draining LNs, it is somewhat surprising that numbers of CD44^{-/-} DCs were almost doubled in the quantified dermis. Perhaps a better way of quantifying cells remaining in the skin would be flow cytometry of enzymatically digested skin (in analogy to the LN analysis).

We apologize for the lack of clarity in our original manuscript and thank the reviewer for alerting us to this. The quantification in Fig 6B (now revised Fig. 7B) refers to the mean fluorescence intensity of BMDCs labelled with either Q-dots 585 or 655, rather than absolute cell numbers, and we have revised the text on page 9 and figure legend accordingly. Also, we show representative images of the quantified tissue whole mounts in the new Supplementary Fig. S7. We chose to analyze BMDC retention in the skin by imaging rather than by enzymatically digesting skin in order to conserve information on the location of retained cells in relation to neighboring dermal lymphatic capillaries. In addition, we were concerned that this approach might not yield quantitative recovery of non-migrating cells, as in our experience, the skin can be more difficult to enzymatically digest than LNs and not all cells are consistently released for analysis.

2. In the DC transmigration assays shown in Figure 7: it would be nice if these findings (involvement of CD44:HA-LYVE-1 axis) could be confirmed by performing transmigration assays with LYVE-1-deficient LECs. According to the hypothesis / findings presented in this study, one would expect that CD44^{-/-} and WT DCs would migrate equally well across LYVE-1-deficient LEC monolayers.

We agree that such an experiment with LYVE-1^{-/-} mLECs might be desirable. However, our attempts to isolate sufficient primary mLECs from LYVE-1-deficient and wild-type mice in parallel, using timed matings to generate the necessary litters, have so far yielded insufficient cell numbers, and the resulting LECs do not readily proliferate beyond 5 passages. Optimizing the procedure will clearly take much further time and this is not helped by current

circumstances with the ongoing COVID19 pandemic. Instead, to address involvement of LYVE-1, we performed transmigration assays of CD44^{+/+} and CD44^{-/-} BMDCs across wild-type mLEC monolayers in the presence of the potent LYVE-1-HA blocking mAb 2125. These new data are now included as an additional panel in Fig. 8F. As the reviewer predicted, disrupting LYVE-1-HA interactions had virtually no effect on the transmigration of CD44^{-/-} BMDC, further supporting the hypothesis that the CD44:HA:LYVE-1 axis is key to DC diapedesis.

3. The authors mention former studies analysing Langerhans cell migration in CD44^{-/-} mice, which apparently yielded conflicting results "due to the abundant expression of CD44 in the tissue". Have the authors performed any FITC painting experiments themselves to address the role of CD44 in the migration of endogenous DCs in this setup (either in complete KOs or in BM-chimeras)? A certain limitation of their study obviously is that it only migration data from adoptively transferred BM-DCs are provided.

The reviewer is correct in stating that our study focuses on BMDCs, although we would like to point out that we also present data from CD44^{-/-} dermal DCs (Fig. 1F, G; Supplementary Fig. S3A). Nevertheless, to satisfy the reviewer's concerns, we now include new data from FITC painting experiments, that address the migration of endogenous dermal DCs (shown in Fig. 6 and Supplementary Fig. S6). These additional studies, in which we assessed migration of such DCs, (defined by expression of CD11c, MHC class II, CD103, EpCAM, Langerin and F4/80) to draining LNs, 24 h after topical application of FITC and oxazolone, indeed showed a significant reduction in their migration in CD44-deficient mice compared to wild-type litter-mate controls.

4. Figure 8/9: While I can understand how anti-CD44 mAb IRAWB14 treatment may enhance DC adhesion, I do not entirely follow why this treatment would lead to reduced DC transmigration and overall migration to dLNs (rather than enhancing this process). I noticed that the control antibody chosen does not bind DCs - a better control in the assay likely would be an antibody that also attaches to the cells (e.g. to CD44 or any other molecule, but without functional consequences).

The data in Figs. 9 and 10 show that enhancing adhesion of BMDCs to mLECs diminishes transmigration *in vitro* and delays trafficking to draining LNs *in vivo*. Receptor-HA interactions are particularly dependent on avidity – a phenomenon that has been termed superselectivity (Dubacheva et al 2015, PMID: 25901321). Hence, surface densities of CD44 sufficient to engage HA chains through multiple bonds can make such binding almost irreversible (Wolny et al 2010 PMID: 20663884). We hypothesize a "Goldilocks" analogy, whereby too much or too little adhesion is detrimental to transmigration. In response to the second issue raised by this reviewer, in figure 3E, we do include such a control antibody, F10-44-2, that binds CD44 but does not influence its function or its level of surface expression.

Minor points:

1. Suppl. 2: please provide FACS quantifications, in addition to the images

Supplementary Fig. 2 now includes representative FACS plots and quantitation for the images.

2. A general remark: Please provide representative FACS plots for all quantifications shown (e.g. Fig. 8A,B, Suppl. Figure 1)

Our revised manuscript includes representative FACS plots for all such data, now shown in revised Fig. 9A, B; Supplementary Fig. S1A; Supplementary Fig. S2B; Supplementary Fig. S5; Supplementary Fig. S6; Supplementary Fig. S7B; and Supplementary Fig. S8.

3. Fig. 6, adoptive transfer experiments: please also show LN cellularity and percentage % data of the DC migration defect

We now show LN cellularity in Figs. 6C, 7F, 10C and percentages in Supplementary Figs. S6B, S7A, S8B and Fig. 10C

4. Figure 9C: The fact that the lymphatic vessel diameter expands upon injection of DCs is surprising and could also be explained by other mechanism than "occlusion of lymph flow due to trapped BM-DCs". E.g. it could be that this antibody induces inflammatory signals in the ear skin (I assume the authors cannot be certain that it remains on the DCs and doesn't bind to other CD44-expressing cells). Please include further possible explanations or else provide more evidence suggesting that lymph flow is indeed reduced.

We have amended the sentence accordingly, to "This is likely due to occlusion of lymph flow by trapped BMDCs, although clearly we cannot rule out the possibility that the IRAWB14 mAb may also evoke bystander inflammatory signals that affect lymph flow" on page 13.

In conclusion, we trust that the revisions and explanations we have provided will satisfy the reviewers' concerns. We believe that the new experiments have strengthened our manuscript and we hope that it will now be acceptable for publication in Life Science Alliance.

Yours sincerely

Louise A. Johnson
Primary corresponding Author
Email: louise.johnson@imm.ox.ac.uk Tel: +44 1865 222414

David G. Jackson
Professor of Human Immunology
Co-Corresponding Author
Email: David.jackson@imm.ox.ac.uk Tel: +44 1865 222313

Rebuttal Figure: Data from replicate experiments
Johnson *et al.*
LSA-2020-00908-T

February 18, 2021

RE: Life Science Alliance Manuscript #LSA-2020-00908-TR

Dr. Louise Anne Johnson
University of Oxford Weatherall Institute of Molecular Medicine, University of Oxford
MRC Human Immunology Unit
John Radcliffe Hospital
Headington
Oxford OX3 9DS
United Kingdom

Dear Dr. Johnson,

Thank you for submitting your revised manuscript entitled "Dendritic cell entry to lymphatic capillaries is orchestrated by CD44 and the hyaluronan glycoalyx". We would be happy to publish your paper in Life Science Alliance pending final revisions necessary to meet our formatting guidelines.

We encourage you to address the concern raised by Reviewer 3, either with changes in the manuscript or with a rebuttal.

Along with the points listed below, please also attend to the following,

- please add callouts for Figures S1 A, B; S2 A, B, and S6 A, B, C to your main manuscript text
- please upload your main manuscript text as an editable doc file
- please use the [10 author names, et al.] format in your references (i.e. limit the author names to the first 10)
- please provide the unedited uncropped source data for the panels in Fig 9E

A. FINAL FILES:

-- High-resolution figure, supplementary figure and video files uploaded as individual files: See our

detailed guidelines for preparing your production-ready images, <https://www.life-science-alliance.org/authors>

B. MANUSCRIPT ORGANIZATION AND FORMATTING:

Sincerely,

Shachi Bhatt, Ph.D.

Executive Editor

Life Science Alliance

<https://www.lsjournal.org/>

Interested in an editorial career? EMBO Solutions is hiring a Scientific Editor to join the international

Reviewer #1 (Comments to the Authors (Required)):

The authors have adequately responded to the criticism.

Reviewer #2 (Comments to the Authors (Required)):

These authors have examined the role of CD44 in the structure of the hyaluronan (HA) glyocalyx surrounding dendritic cells and entry of these cells into lymphatic capillaries. They report that a) CD44 dynamically controls the density of the hyaluronan glyocalyx, b) CD44 is required for DC adhesion and transmigration, and c) that these pericellular CD44/HA interactions regulate the efficiency of DC trafficking to lymph nodes. These points are now addressed in the manuscript.

Reviewer #3 (Comments to the Authors (Required)):

The authors have performed several major experiments and successfully revised the manuscript to address my comments. E.g., they have performed FITC painting experiments that confirm the contribution of CD44 to the migration of endogenous DCs. Moreover, they have performed additional in vitro transmigration assays with WT and CD44^{-/-} BM-DCs in presence / absence of a LYVE-1 blocking antibody. These experiments show that LYVE-1 blockade only reduces the transmigration of WT but not of CD44-deficient DCs (Fig. 8F), thereby providing further support for their hypothesis of CD44:HA:LYVE-1 axis mediating DC transmigration across lymphatic endothelium. The authors have also performed further experiments with an additional control antibody, which binds CD44 but does not modulate its function (Fig. 3E). Moreover, they now show representative FACS gatings of all major experiments in the Supplement. Last but not least, they have revised the manuscript text to verbally address some further comments I had.

I only have one final remark to make, concerning the authors' reply to my "major point 1", which in my opinion was not addressed to completion: Although the authors now better explain and illustrate (with images) what they measured in Figure 7B, I am rather puzzled by these findings and their interpretation (i.e. an increase in fluorescent intensity at the site of dermal injection, in the case of CD44^{-/-} as compared to WT DC adoptive transfer). As already pointed out in my question and as seen in Fig. 7A: In adoptive transfer studies, typically less than 1% of all injected DCs (according to the Material and Methods section, the authors transferred 2 million DCs) typically arrive in the draining lymph node. This means that approx. 95% (perhaps even more!) of all DCs should remain at the injection site (where most of them will die). Mathematically and experimentally, I cannot follow how one should be able to measure an increase in signal at the injection site, if only 1-5% of the cells disappeared from that site. I would recommend to omit these data from the manuscript. In my opinion, the results shown in Fig. 7A are very clear and strong and stand for themselves. But I leave this decision up to authors.

Overall, I congratulate the authors on a very interesting and thoroughly conducted study!

February 22, 2021

RE: Life Science Alliance Manuscript #LSA-2020-00908-TRR

Dr. Louise Anne Johnson
University of Oxford Weatherall Institute of Molecular Medicine, University of Oxford
MRC Human Immunology Unit
John Radcliffe Hospital
Headington
Oxford OX3 9DS
United Kingdom

Dear Dr. Johnson,

Thank you for submitting your Research Article entitled "Dendritic cell entry to lymphatic capillaries is orchestrated by CD44 and the hyaluronan glycoalyx". It is a pleasure to let you know that your manuscript is now accepted for publication in Life Science Alliance. Congratulations on this interesting work.

*****IMPORTANT:** If you will be unreachable at any time, please provide us with the email address of an alternate author. Failure to respond to routine queries may lead to unavoidable delays in publication.*******

DISTRIBUTION OF MATERIALS:

Again, congratulations on a very nice paper. I hope you found the review process to be constructive and are pleased with how the manuscript was handled editorially. We look forward to future exciting

submissions from your lab.

Sincerely,

Shachi Bhatt, Ph.D.

Executive Editor

Life Science Alliance

<https://www.lsjournal.org/>

Interested in an editorial career? EMBO Solutions is hiring a Scientific Editor to join the international Life Science Alliance team. Find out more here -

https://www.embo.org/documents/jobs/Vacancy_Notice_Scientific_editor_LSA.pdf